# Nicotinamide metabolism is essential for Hepatitis C Virus replication and the production of infectious Lipo-Viro-Particles

Johan Toesca[1], Marion Castell[1], Clémence Jacquemin[1], Alexandre Lalande[1,2], Eva Ogire[3], Julien Burlaud-Gaillard[4], Philippe Roingeard[4], Christophe Ramière[1,5], Cyrille Mathieu[3], Laure Perrin-Cocon[1], Vincent Lotteau[1,2], Pierre-Olivier Vidalain[1], Olivier Diaz [1]*

1 CIRI, Centre International de Recherche en Infectiologie, Team Viral Infection, Metabolism and Immunity, Univ Lyon, Inserm, U1111, CNRS, UMR5308, Université Claude Bernard Lyon 1, Ecole Normale Supérieure de Lyon, Lyon, France, 2 Laboratoire P4 INSERM-Jean Mérieux, Lyon, France, 3 CIRI, Centre International de Recherche en Infectiologie, Team NeuroInvasion, Tropism and Viral Encephalitis, Univ Lyon, Inserm, U1111, CNRS, UMR5308, Université Claude Bernard Lyon 1, Ecole Normale Supérieure de Lyon, Lyon, France, 4 Inserm, U1259, Morphogénèse et Antigénicité du VIH, des Virus des Hépatites et émergents (MAVIVHe) & Inserm, Analyse des Systèmes Biologiques (ASB), Université de Tours et CHRU de Tours, Tours, France, 5 Virology Department, Institut des Agents Infectieux, Hôpital de la Croix Rousse, Hospices Civils de Lyon, Lyon, France

* olivier.diaz@inserm.fr

## Abstract

### Background & Aims

Hepatitis C virus (HCV) has the unique characteristic of forming lipo-viro-particles (LVPs), which are lipid-rich virions containing both the viral components and host apolipoproteins such as ApoB and E. This unique composition gives to LVPs a low buoyant density, facilitates their entry into the hepatocyte, and is a hallmark of highly-infectious HCV particles. Although recent studies have shown that inhibiting NAD biosynthesis can both disrupt central carbon metabolism and thereby interfere with the replication of hepatotropic viruses such as dengue virus (DENV) and hepatitis B virus (HBV), the impact of nicotinamide biosynthesis inhibition on HCV replication and LVP formation has not yet been explored.

### Methods

We therefore investigated the dependance of HCV on NAD(H) biosynthesis in Huh7 cells by using the antimetabolite 6-Aminonicotinamide (6-AN) or by specifically inhibiting NAMPT, a key enzyme in the nicotinamide salvage pathway. The impact on cellular metabolism was assessed by LC-MS/MS to quantify metabolites, by confocal microscopy to analyze lipid droplets and by ELISA for ApoB/E secretion. Glycolytic activity and mitochondrial respiration were evaluated by real-time measurement of extracellular acidification rate (ECAR) and oxygen consumption rate (OCR), respectively. Consequences on viral replication were analyzed using both a

**Data availability statement:** All relevant data are within the manuscript and its Supporting information files.

**Funding:** This research was funded by the Institut National de la Santé et la Recherche Médicale (INSERM), the Centre National de Recherche Scientifique (CNRS), Université Claude Bernard Lyon I (UCBL) and the Agence Nationale de Recherches sur le Sida et les hépatites virales (ANRS|Maladies infectieuses émergentes) grant number ECTZ244976 to O.D. Ph.D fellowships for J.T. (ECTZ317854) and M.C. (ECTZ246534) were granted by ANRS. We thank the Agence National de la recherche (ANR) grant number ANR-21-MATP-0701 to OD. We thank the Fondation CNRS, project VIRIMI, for its financial support. The funders had no role in study design, data collection and analysis, decision to publish, or preparation of the manuscript.

**Competing interests:** The authors have declared that no competing interests exist.

subgenomic replicon (strain JFH1) and the full-length infectious virus (strain Jc1). The effect of 6-AN on the formation of double-membrane vesicles (DMVs) where the virus replicates was determined by transmission electron microscopy. Finally, the secretion and specific infectivity of virions were analyzed by RT-qPCR and titration technics, either before or after separation by density-gradient centrifugation to focus on LVPs.

## Results

Pharmacological inhibition of NAD(H) biosynthesis in Huh7 cells impaired HCV replication, the formation of DMVs and the production of infectious LVPs. Mechanistically, 6-AN drastically inhibited glycolysis but increased oxidative phosphorylation as compensatory mechanism. This metabolic reprogramming was associated with decreased intracellular levels of triglycerides, smaller lipid droplets and reduced secretion of Apo B and E, which altogether could explain the impact of 6-AN on HCV replication and the production of LVPs.

## Conclusions

Inhibiting NAD(H) biosynthesis disrupts central carbon metabolism, reduces intracellular triglycerides and blocks ApoB+-lipoprotein secretion—a pathway essential for HCV replication and LVP production. These results reveal, for the first time, that HCV life cycle is critically dependent on NAD(H) metabolism, reinforcing the interest of this pathway as a potential therapeutic target against hepatotropic viruses.

## Author summary

Hepatitis C virus (HCV) produces highly infectious particles known as lipo-viro-particles (LVPs), which are characterized by an unusually high lipid content. The formation of these particles relies on the host cell's lipid metabolism, but the metabolic pathways that support this process are still incompletely understood. In this study, we show that HCV critically depends on cellular nicotinamide adenine dinucleotide (NAD(H)) biosynthesis, a central metabolic pathway involved in energy production. By pharmacologically inhibiting NAD(H) metabolism in liver cells, we disrupted key metabolic processes, including glycolysis and lipid storage. This metabolic reprogramming reduced the formation of intracellular membranes required for viral replication, decreased intracellular neutral lipids, and impaired the secretion of apolipoproteins essential for LVP assembly. As a result, HCV replication and the production of infectious LVPs were strongly reduced. Our findings reveal an unexpected link between NAD(H) metabolism and the HCV life cycle, highlighting host metabolic pathways as potential targets for the development of new antiviral strategies against hepatotropic viruses.

## Introduction

Viruses are obligate parasites that rely on the biosynthetic machinery of the infected cell for their replication. Over the course of their evolution, viruses have selected various strategies to manipulate the host cell's metabolism, thus ensuring an adequate supply of energy and metabolites. This is crucial for the formation of viral factories, where viral replication takes place, and to support the synthesis of biomolecules required for the production of new virions. Viruses can stimulate both anabolic and catabolic processes to meet their needs and create an environment that is conducive to viral replication. We are just beginning to understand how viruses modulate metabolic pathways and which key cellular enzymes are hijacked during infection [1]. In this context, viruses often induce a metabolic phenotype that is characterized by an increased glycolysis and a decreased oxidative phosphorylation to promote metabolite biosynthesis. By inducing such a phenotype, viruses promote the synthesis of biomolecules they need for their replication, such as nucleotides and lipids. For instance, infections caused by herpesviruses [2], Coxsackievirus [3], influenza virus [4], human immunodeficiency virus (HIV) [5], hepatitis C virus (HCV) [6,7] or dengue virus (DENV) [8] have been associated with upregulation of glycolysis. Other viruses such as herpes simplex virus (HSV), vaccinia virus [9] and adenovirus infection trigger an increase in glutaminolysis. Interestingly, the induction of lipogenesis has been established as critically required for the replication of *Flaviviridae*, such as HCV, DENV, Zika virus (ZIKV) or West Nile virus (WNV) [7,10–13]. This suggests that, although there is no common control mechanism, there is a shared dependence on lipogenesis among flaviviruses.

Nicotinamide adenine dinucleotide (NAD) is a cellular coenzyme involved in numerous redox reactions. In particular, the oxidized form of NAD ($NAD^+$) is reduced to NADH during glycolysis, in the tricarboxylic acid (TCA) cycle, and during oxidation of fatty acids. NADH is then oxidized back to $NAD^+$ by transferring electrons to the electron transport chain of mitochondria. Additionally, $NAD^+$ acts as a substrate for several enzymes, including $NAD^+$-dependent deacetylases (sirtuins), poly (ADP-ribose) polymerases (PARPs), $NAD^+$ glycohydrolase (CD38, SARM1) and cyclic ADP-ribose (cADPR) synthase. These enzymes play key roles in essential physiological processes such as energy metabolism, gene transcription, epigenetic regulation, DNA repair and cellular senescence, processes that can also be modulated during infection. Interestingly, recent studies have established the antiviral effect on ZIKV, HBV and DENV of drugs inhibiting NAD(H) metabolism [8,14,15]. Previous studies, including our own work, have described the induction of cellular glycolysis and lipogenesis during HCV infection and have established the dependence of HCV to these metabolic pathways for replication [6,7,16]. We hypothesized that HCV replication in hepatocytes might also be dependent on NAD(H) metabolism as previously reported for DENV and ZIKV.

HCV is a single-stranded positive RNA virus, which encodes 3 structural proteins (*i.e.,* core, E1 and E2 glycoproteins) and 7 non-structural proteins (i.e., p7, NS2, NS3, NS4A, NS4B, NS5A and NS5B) [17]. During infection, non-structural proteins induce the biosynthesis of intracellular structures called membranous web, forming double-membrane vesicles (DMVs). These DMVs serve as the sites of viral genome replication and are closely associated with downstream steps of virion assembly and lipidic envelopment [17–19]. This final step involves the subversion of the hepatic lipoprotein synthesis pathway to produce Lipo-Viral-Particles (LVPs), a subpopulation of virions that are enriched in triglycerides (TG) [18,19]. These LVPs are hybrids made of both capsid and viral glycoproteins as well as apolipoproteins and neutral lipids that are normally associated to very-low-density lipoproteins (VLDL) [20,21]. Consequently, viral particles are heterogeneous in terms of density, with the lowest density particles being the most infectious. The induction of lipogenesis in the HCV-infected hepatocyte has been reported in multiple studies, supporting the synthesis of phospholipids and triglycerides that is necessary for viral-induced DMV and LVP formation [22]. When the activity of acetyl-CoA carboxylase, a key enzyme in lipid biosynthesis, is inhibited, the formation of replication structures is impaired [23]. Additionally, inhibiting VLDL assembly by targeting the microsomal triglyceride transfer protein (MTTP), an enzyme essential to lipoprotein assembly, has been shown to decrease LVP formation [24]. Even though several mechanisms by which HCV controls hepatocyte metabolism have been described, the effect on HCV replication of antimetabolites targeting NAD(H) biosynthesis has never been reported yet. In this study, we describe how inhibition of NAD(H) metabolism, in particular inhibition

of the salvage-pathway, impairs HCV replication in hepatocytes. We have notably established that targeting this host metabolic pathway prevents the formation of DMVs and reduces intracellular TG contained in lipid droplets, which are important for virion assembly and the production of LVPs.

## Materials and methods

### Cells and reagents

Huh7 and Huh7.5 cells (a gift from Marco Binder lab; Heidelberg University; Germany) were grown in DMEM, high glucose, with GlutaMAX (Gibco; France) supplemented with 10% fetal calf serum (FCS; Biosera; France) and 100 IU/mL penicillin/streptomycin (Gibco; France). Cells were grown at 37°C and 5% $CO_2$. 6-Aminonicotinamide (6-AN), Nicotinamide (NAM), Nicotinamide Riboside (NR), FK866, STF-118804 were obtained from MedChemExpress (CliniSciences; France).

### Human hepatocyte derived from chimeric mice (HepaSH) cultures

Freshly purified cells were obtained from Biopredic International, seeded upon reception in glycogen-coated well plates as recommended by the supplier and cultured for 4 days in William-E medium (Gibco 12551032) containing 100μg/mL Penicillin/Streptomycin, 2mM L-Glutamine, 1% Non-Essential Amino Acids, Hydrocortisone, 5μg/mL Insulin, 2ng/mL EGF, 5% SVF, 2% DMSO and 10mM HEPES, before infection.

### *In vitro* RNA transcription

Plasmids containing the subgenomic replicon HCV-JFH1-R2A or HCV-Jc1 full length were obtained from Prof. Ralf Bartenschlager (Department of Molecular Virology, Heidelberg, Germany) and RNA synthetized as previously described [25]. Briefly, plasmids containing the subgenomic replicon and the plasmid containing Jc1 strain were linearized with XbaI and AseI enzymes, respectively. Linearized DNA was purified by phenol/chloroform extraction, precipitated with isopropanol and resuspended in RNase-free water. In vitro transcription was performed with T7 RNA polymerase (T7 Express Large Scale RNA Production System, Promega). RNA was extracted with acidic phenol and chloroform, precipitated with isopropanol and resuspended in RNase-free water.

### Electroporation of *in vitro*-transcribed RNA

HCV-RNA was electroporated into Huh7 cells as previously described [26]. Briefly, cells were trypsinized and washed twice with PBS. $4 \times 10^6$ cells were resuspended in 400 μL of cytomix (120 mM KCl, 150 mM $CaCl_2$, 10 mM $K_2HPO_4$/$KH_2PO_4$, 25 mM HEPES, 2 mM EDTA, 5 mM $MgCl_2$, pH 7.6) and electroporated in presence of 5 μg of HCV-RNA with the following conditions: 975 μF and 270 V with a Gene Pulser system (Bio-Rad) in a cuvette with a gap width of 0.4 cm (Bio-Rad). Immediately after electroporation, cells were resuspended in 20 mL of complete medium and seeded in culture plates suitable for subsequent experiments.

### Luciferase assay

When electroporated with subgenomic luciferase-replicon, cells were seeded in white opaque 96-well microplates (200 μL of cell suspension per well). 72 h post-electroporation, 100 μL of medium were removed and luciferase activity was assessed by addition of 50 μL of Luciferase Assay System mix (Promega). After 10 min incubation, luminescence was quantified using a Tristar5 luminometer (Berthold, Freiburg, Germany) for 1s, and expressed as relative light units (RLU). Luciferase activity from cells harvested 4 h after electroporation was used to determine transfection efficiency.

## Virus stock production and titration

Jc1 and Jc1-E2-FLAG virus stocks were generated as previously described [6]. Briefly, master stocks were generated by electroporation of Huh7.5 cells with *in vitro* transcripts of the full-length HCV genomes as described above. Immediately after electroporation cells were resuspended in 12 mL of complete medium and seeded in two 10 cm diameter dishes. 6 h post-electroporation medium was replaced with fresh media. 72 h post-electroporation supernatants were harvested, clarified through a 0.45 µm filter and stocked at -80°C before titration. A second harvesting of the culture was performed 72 h later and treated identically. For titration, Huh7.5 cells were seeded in 48-well plates at $2x10^4$ cells/well 24 h before their infection with serial dilutions of virus-containing culture supernatants. At 72 h post-infection, cells were fixed with ice-cold ethanol for 30 min before staining with anti-NS5A antibody (clone 9E10, Sigma-Aldrich) combined with anti-mouse IgG HRP-conjugate antibody.

## dsRNA foci staining and quantification

Huh7 cells were seeded in ibidi µ-Slide 8-well chambers 24 h before infection (MOI 1) and treated or not with 100 µM 6-AN±500 µM NAM for 72 h. After removing the culture supernatant, cells were washed with PBS and fixed for 15 min at room temperature with 4% paraformaldehyde. They were permeabilized for 30 min at 4 °C using PBS+0.2 M glycine+1% Triton X-100, then blocked for 30 min at room temperature with PBS+0.2 M glycine+0.5% BSA. After a PBS+0.2 M glycine wash, cells were incubated for 1 h at room temperature with anti-dsRNA antibody (1:100, clone J2, Jena Bioscience RNT-SCI-10010200) diluted in PBS+0.2 M glycine. Cells were washed twice with PBS+0.2 M glycine, incubated with Alexa Fluor 488-conjugated goat anti-mouse IgG (1:2000, Invitrogen A-11029) for 1 h at room temperature, washed again, stained for 5 min with 5 µM Hoechst in PBS, washed, and imaged on a Yokogawa HCS CQ1 confocal system. For dsRNA foci quantification, Z-stacks (11 µm total, 1.8 µm steps) were acquired in random fields (14 fields per condition from two independent slides) using a UPLSAPO40X2 objective (Olympus). Acquisition settings were 100% excitation and 750 ms exposure for the 488 nm channel, and 50% excitation and 500 ms exposure for the 405 nm channel. Images were analyzed in the CQ1 software using the "Dots in Cell body" template in slice mode to automatically segment dsRNA foci. Nuclei segmentation used the following steps: ThresholdToZero (400), Threshold (450), ErosionCircle (0.7 µm), DivideEachRegion, SizeFilter (450–20000 µm³), FillUp, ExcludeEdge. Cell bodies were segmented from green-channel autofluorescence using: MeanImage (5 µm mask), Threshold (150), DivideEachRegion, ExpandRegion3D (nucleus-guided), SizeFilter (20–10,000,000 µm³). dsRNA dots were segmented using: Gaussian filter (0.1 µm mask), Threshold (510), FindMaximumImage (remove size <0.3 µm), DivideEachRegion, ExpandRegion3D (Gaussian input; Threshold 450, 5 iterations), SizeFilter (0.1–1,000,000,000), and DilateRegion (0.1 µm). Low-magnification single-plane images were acquired using a UPLSAPO4X objective (Olympus) with the 488 nm laser (100% excitation, 2000 ms exposure), and stitched with the microscope software to generate mosaics covering nearly the entire well area.

## Iodixanol density gradient

10 mL of supernatant from HCV-infected cells were concentrated 10 times on a 100,000 MWCO PES membrane Vivaspin (Sartorius). 1 ml of the resulting concentrated supernatant was layered on the top of a 0–30% continuous preformed iodixanol gradient (Optiprep; Merck). Gradients were centrifuged for 16 h at 32,000 rpm in a SW41 swinging rotor at 4°C using an Optima L-90 K Beckmann centrifuge. Fifteen fractions of 750 µl were collected from the top and were analyzed for virus titer and viral RNA content.

## Apolipoprotein B (ApoB), Apolipoprotein E (ApoE) and triglycerides quantifications

ApoB and ApoE concentrations in medium and gradients fractions were determined by ELISA as previously described [27], using the following antibodies: anti-human ApoB mAbs (LDL 20/17), anti-human ApoB mAb (LDL 11) biotin,

anti-human ApoE mAb (E276), anti-human ApoE mAb (E887) biotin (MABTECH, Sweden). Triglycerides (TG) were quantified using a specific enzymatic assay (Millipore Sigma-Aldrich).

## Intracellular lipid droplet staining

Supernatant of cultures were removed and cells were washed with PBS before fixation with a 4% formaldehyde solution during 15 min at RT. Cells were then washed twice with deionized water before a 5 min incubation with isopropanol 60%. Isopropanol was removed and Oil-Red-O solution (Millipore Sigma-Aldrich) added on cells for 15 min at RT. The Oil-Red-O working solution was prepared extemporaneously by mixing 6 ml of a 0.3% Oil-Red-O w/v isopropanol solution with 4 ml of water. Cells were then extensively washed with water to remove the exceeding dye before nucleus counter-staining with NucBlue Fixed Cell Stain ReadyProbes reagent (ThermoFisher Scientific) and observation with an inverted confocal microscope Zeiss LSM800 (x63 objective). The intracellular lipid droplet contents were analyzed using the IMARIS software.

## Quantification of viral RNA and gene expression analysis by RT-qPCR

RNAs were extracted either from cells or supernatants with TRI Reagent (ThermoFischer Scientific), reverse transcribed with High-Capacity RNA-to-cDNA kit (Applied Biosystems), and HCV RNAs were quantified using SYBR Green PCR kit QuantiNova (Qiagen) on an Applied StepOne Real-Time PCR apparatus using the forward primer 5'-TCTGCGGAACCG GTGAGTA-3' and the reverse primer 5'-TCAGGCAGTACCACAAGGC-3'. For quantification of cellular gene expression, the specific following primers were used: NAMPT (forward primer 5'-GGTTCTGGTGGAGGTTTGCT-3' and the reverse primer 5'-CTGCTGGCGTCCTATGTAAAGA-3'), NAPRT (forward primer 5'-AGCCACGAATGAAGCTGACCGA-3' and the reverse primer 5'-CACTGGCTCTTCTGCTAACTGC-3') and QPRT (forward primer 5'- CTCCAGTGCCCAAAATCCAC-3' and the reverse primer 5'- CTGACCCTAAAGATGTGTGACC-3'). Expression of RPL13A was used as housekeeping gene (forward primer 5'-AAAAGCGGATGGTGGTTCCT-3' and the reverse primer 5'-GCTGTCACTGCCTGGTACTT-3').

## Metabolomic analysis

Metabolites extractions and metabolomic analysis were performed on four independent biological replicates. For each condition $2x10^6$ Huh7 cells were seeded in 10 mm dishes and grown for 24 h in 10 mL of culture medium. Supernatant was removed and replaced by 8 mL of fresh culture medium containing or not 6-AN at the final concentration of 100 µM for 48h. Culture medium was removed and metabolites were immediately extracted from the cell monolayer in 2 mL of ice-cold (-20° C) 80% MS-grade methanol (Sigma-Aldrich) diluted with sterile pyrogen free water Otec (Aguettant; France). Cell extracts were immediately transferred into tubes and vortexed for 1 min, samples were stored at -80°C. Sample analysis was carried out by MS-Omics (Denmark) as follows. Samples were dried under nitrogen flow and reconstituted in 140 µl MQW. After reconstitution samples were filtered and additionally diluted 10 times in eluent A for semi-polar metabolites analysis and 5 times in eluent A for polar metabolites analysis. The analyses were carried out using a Thermo Scientific Vanquish LC coupled with a high-resolution quadrupole-orbitrap mass spectrometer (Q Exactive HF Hybrid Quadrupole-Orbitrap, Thermo Fisher Scientific). An electrospray ionization interface was used as ionization source. Analysis was performed in negative and positive ionization mode. For semi-polar metabolites, the UPLC was performed using a slightly modified version of the protocol described by C.E. Doneanu et al. [28]. For polar metabolites, the UPLC was performed using a slightly modified version of the protocol described by Hsiao et al. 2018 [29]. Peak areas were extracted using Compound Discoverer 3.2 (Thermo Fisher Scientific) and Skyline [30]. Identification of compounds were performed at four levels; Level 1: identification by retention times (compared against in-house authentic standards), accurate mass (with an accepted deviation of 3ppm), and MS/MS spectra, Level 2a: identification by retention times (compared against in-house authentic standards), accurate mass (with an accepted deviation of 3ppm). Level 2b: identification by accurate mass (with

an accepted deviation of 3ppm), and MS/MS spectra, Level 3: identification by accurate mass alone (with an accepted deviation of 3ppm). A total of 1,768 compounds were detected in the samples. Hereof, were 522 annotated on level 3, 55 on level 2b, 68 on level 2a, and 83 on level 1. Compounds analyzed in this study are from Level 1 or 2a. The results from the analyses are presented as log2 values of the ratio between the average of 6-AN-treated vs control samples. Statistical significance was determined with a Student's t-test for paired samples (two-tailed). MetaboAnalyst (version 6.0) was used for metabolite enrichment analyses.

### Real-time monitoring of metabolic phenotype with a Seahorse XF analyzer

Cells were seeded in Seahorse XF 96-well microplates (Agilent), coated with Poly-L-Lysine 0.01% (Sigma-Aldrich), at $8\times10^3$ cells/well in 200 µL of DMEM medium (Gibco) supplemented with 10% FCS (Biosera), 2 mM L-glutamine, 100 U/mL penicillin/streptomycin, treated with 100 µM 6-AN, 500 µM NAM or $H_2O$ alone and incubated at 37°C and 5% $CO_2$ for 72 h. The assay was initiated by replacing growth medium with prewarmed Seahorse assay medium (XF DMEM pH7.4; 103575–100; Agilent) supplemented with glucose (10 mM) and L-glutamine (2 mM). Cells were washed with 200 µL assay medium and incubated at 37°C for 1 h without $CO_2$. The medium was replaced by 180 µL of prewarmed assay medium prior measurement of oxygen consumption rate (OCR) and extracellular acidification rate (ECAR) with the Seahorse XFe24 analyzer using the XF Cell Mito Stress Test (Agilent) or XF Glycolytic Rate assay (Agilent). The number of cells was determined at the end of the run after Hoechst 33342 staining and cell counting using Cytation 1 cell imaging reader (Agilent BioTek). Results were normalized by cell count and analyzed using the Seahorse Wave software.

### Quantification of cell proliferation

Cellular amounts in culture wells were determined with the CellTiter-Glo Luminescent Cell Viability Assay (G7570; Promega; France), which relies on ATP quantification as a proxy for cell number in a well, or by staining cell nuclei with Hoechst followed by quantification of the fluorescent signal. Huh7 cells were seeded at $8\times10^3$ cells/well in quintuplicates in white 96-well plates for the CellTiter-Glo detection or black plates with clear bottoms for fluorescence-based cell quantification. Cells were incubated for 96 h in 200 µL of culture medium with DMSO alone or indicated drugs (6-AN at 100 µM or NAM at 500 µM). For ATP quantification in culture wells, 100 µL of culture medium were removed first, and 50 µL of CellTiter-Glo reagent (Promega) were added in each well. After 10 min of incubation at RT, luminescence was quantified with Tristar 5 Multimode reader (Berthold; Germany). For staining cell nuclei, Hoechst 33342 (Thermo Scientific) was added to each well at 40 µM and incubated for 30 min at 37°C. Wells were washed with PBS and fluorescence (excitation 350 nm/ emission 461 nm) was measured with a Tristar 5 Multimode reader.

### Cell viability

Cytotoxicity was evaluated by CellTox Green cytotoxicity assay (Promega). $8\times10^3$ cells/well were seeded in black 96-well plates and cultured for 72 h in presence of 6-AN (100 µM), 6-AN (100 µM) plus NAM (500 µM), or DMSO alone. Then, cells were labeled with 1/500 dilution of CellTox green dye during 15 min at room temperature shielded from ambient light, before fluorescence measurement at 490 nm (excitation) and 520 nm (emission), using a TECAN M200 microplate reader. Under the same culture conditions, a series of wells were treated with 10% Triton X-100 one hour before the cytotoxicity assay to determine the maximal fluorescent signal corresponding to 100% toxicity for each condition.

### Double membrane vesicles assessment by electron microscopy

$5\times10^5$ of Huh7 cells were seeded on T75 flasks 24 h before infection with HCV Jc1 at a MOI of 1. 6 h after inoculation, culture medium was removed and replaced by fresh medium containing 6-AN (100 µM), 6-AN (100 µM) plus NAM (500 µM), or DMSO alone. After 72 h of culture, cells were harvested, washed twice with PBS before being fixed by incubation for

48h in 4% paraformaldehyde and 1% glutaraldehyde in 0.1 M phosphate buffer pH 7.2 and then washed twice with PBS. Cells were then incubated with 2% osmium tetroxide (Agar Scientific, Stansted, UK) for 1 h. Fixed cells were then fully dehydrated in increasing concentration of ethanol (70%, 90% and 100%) and then propylene oxide (100%). Fixed cells were then impregnated with a 1:1 mixture of propylene oxide/Epon resin (Sigma) and incubated overnight in pure resin. Cells were then embedded in Epon resin (Sigma) and left to polymerize for 48 h at 60°C. Ultrathin sections (80 nm) were cut with an EM UC7 ultramicrotome (Leica Microsystems, Wetzlar, Germany). Contrast staining was performed with 2% uranyl acetate (Agar Scientific) and 5% lead citrate (Sigma), and the samples were then observed with a JEOL JEM-1011 (Tokyo, Japan) transmission electron microscope operated at 100 kV and equipped with an Ametek-GATAN RIO9 CMOS camera.

**Bioinformatics and statistical analyses:** Significance values were calculated by applying tests indicated in the Figure legends using the GraphPad Prism 10 software (GraphPad Software, USA). P values under 0.05 were considered statistically significant and are indicated within the Figures.

## Results

### HCV replication and DMV formation depend on NAD(H)

In mammalian cells, NAD(H) and NADP(H) are synthesized *de novo* either from tryptophan via the kynurenine pathway or from nicotinic acid (NA) via the Preiss-Handler pathway (Fig 1A). NAD(H) is also produced via the salvage pathway using nicotinamide (NAM), a byproduct of NAD(H)-consuming reactions (Fig 1A). 6-aminonicotinamide (6-AN) is an analogue of NAM which competes with NAM as a substrate for nicotinamide phosphoribosyl transferase (NAMPT), the first enzyme of the salvage pathway. The use of 6-AN leads to the formation of 6-aminonicotinamide adenine dinucleotide (6-ANAD) and 6-ANAD phosphate (6-ANADP), which are non-reducible analogues of NAD(H) and NADP(H). The consequence is an inhibition of enzymes using NAD(H) and NADP(H) in redox reactions. Since 6-AN was previously reported to inhibit the replication of vaccinia virus, ZIKV, HBV, and DENV [8,14,15,31], we sought to determine whether this compound could also inhibit HCV replication. Huh7 cells were transfected with a subgenomic replicon (strain JFH1) expressing luciferase as a reporter instead of structural proteins, and then treated with increasing doses of 6-AN for 72 h. As shown in Fig 1B, HCV replication was inhibited in a dose-dependent manner, by 80% at 100 µM and 90% at 500 µM. To assess the viability of cells treated with 6-AN for 72 h, we measured intracellular ATP levels, performed cell count under treatment and conducted a cytotoxicity assay (S1 Fig). As described elsewhere [14], we observed that 6-AN treatment reduced intracellular ATP level, indicating a decreased metabolic activity (S1A Fig). We also observed a decreased cell count by ~30% at 100 µM of 6-AN (S1B Fig), but no impact on cellular integrity in a membrane integrity assay using CellTox Green dye (S1C Fig) showing a delay in cell growth. Thus, 6-AN effectively inhibits HCV replication in Huh7 cells, likely by limiting cellular metabolic activity and in particular ATP availability among other metabolites, while not affecting cellular membrane integrity. This shows that, without killing cells, 6-AN induces a metabolic slowdown limiting viral replication. Interestingly, viral replication was restored in a dose-dependent manner when the culture medium was supplemented with an excess of NAM or nicotinamide riboside (NR), two NAD(H) precursors (Fig 1C and 1D).

As a proxy of viral replication, we investigated the impact of 6-AN on viral replication complexes. We thus infected Huh7 cells with HCV (Jc1 strain) and treated cells for 72 h with either 100 µM 6-AN or 6-AN supplemented with 500 µM NAM. Replication foci were labeled using an anti-dsRNA antibody and visualized with an Alexa Fluor 488–conjugated secondary antibody, while nuclei were counterstained with Hoechst (Fig 1E–1G). Confocal microscopy analysis revealed that the percentage of cells exhibiting detectable replication complexes was significantly reduced following 6-AN treatment, indicating impaired viral spread upon nicotinamide metabolism inhibition (Fig 1F). This effect was reversed by NAM supplementation, which restored the proportion of cells displaying replication foci. Interestingly, the number of replication foci per infected cell was not markedly altered by 6-AN treatment. However, analysis of individual foci volumes showed that they were, on average, smaller in 6-AN–treated cells. These results suggest that while the number of replication complexes per

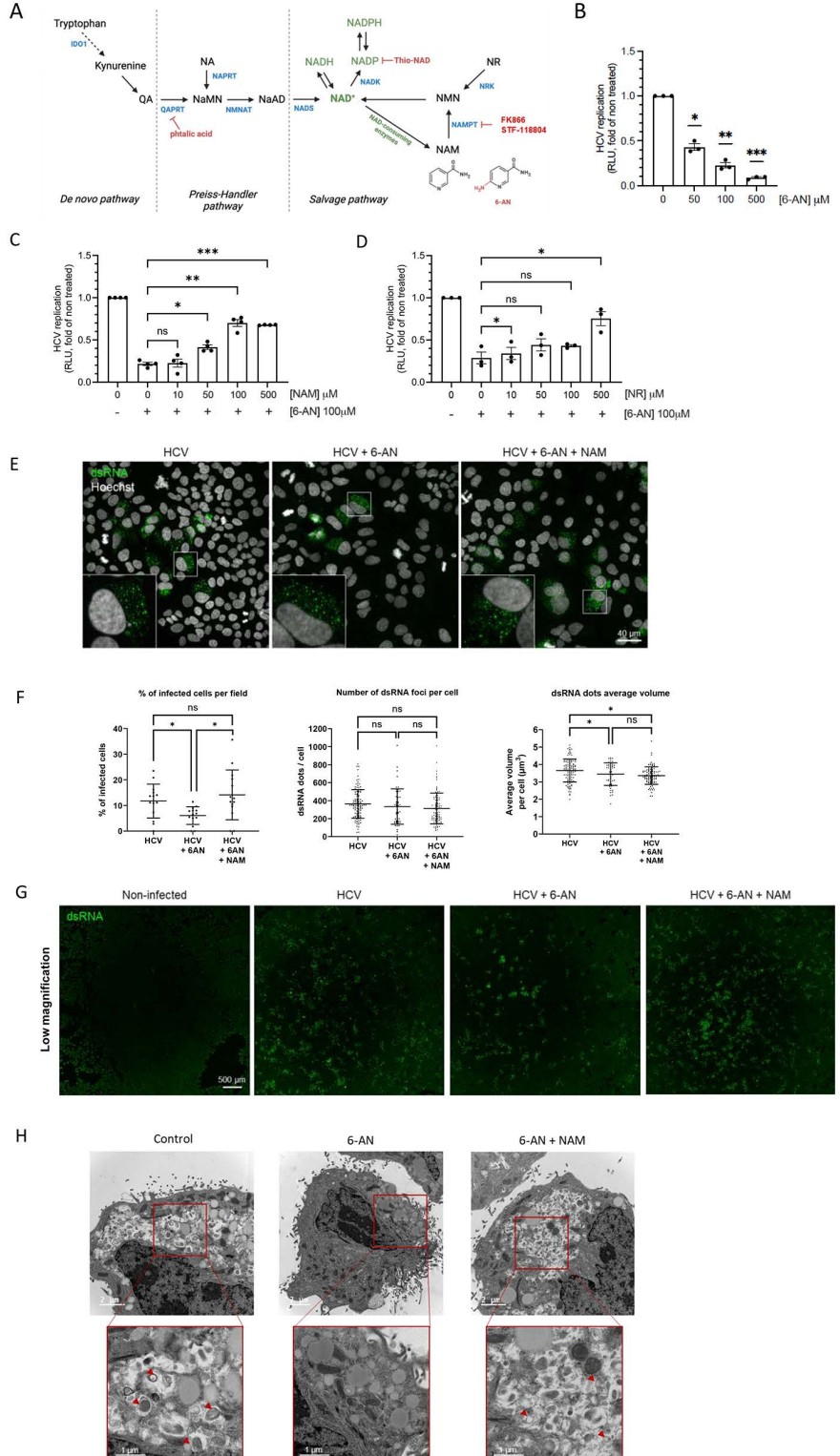

**Fig 1. HCV replication depends on NAD(H).** **(A)** Schematic representation of NAD(H), NADP(H) biosynthesis pathways and targets of drugs used in the study. **(B-D)** Huh7 cells were electroporated with subgenomic replicon and cultured for 72 h with the indicated treatment. Then, Renilla luciferase activity was measured in cell homogenates and viral replication normalized to non-treated control (0 or Ctl.). Are presented means±SEM.

**(B)** Replicon-electroporated Huh7 cells were cultured in presence of increasing concentrations of 6-AN (0, 50, 100 and 500 µM), n = 3, one-way ANOVA, Bonferroni-Sidák adjusted p-value for multiple comparison. **(C)** Cells were cultured with or without 100 µM of 6-AN and increasing doses of NAM, n = 4, one-way ANOVA, Bonferroni-Sidák adjusted p-value for multiple comparison to 6-AN treatment. **(D)** Cells were cultured with or without 100 µM 6-AN and increasing doses of nicotinamide riboside (NR), n = 3, one sample t-test, Bonferroni-Sidák adjusted p-value for multiple comparison to condition control. Are presented means ± SEM (n.s. non significative, *p < 0.01, **p < 0.01, ***p < 0.001). **(E)** Huh7 cells were infected and treated or not with 6-AN ± NAM, stained for dsRNA and imaged by confocal microscopy. Representative Z-stack maximum intensity projections of cells stained for dsRNA. Closeups and whole fields of view (with stronger brightness and contrast adjustment to facilitate the visualization of foci-containing cells) are shown. **(F)** Corresponding quantifications of the proportion of infected cells (dsRNA foci-containing cells), dsRNA foci abundance per cell and dsRNA foci volume. Data are presented as mean ± SD, and were analyzed using one way ANOVA, *p < 0.05, ns none significative. **(G)** Mosaic microscopy images showing wide field of views of the cell monolayers stained for dsRNA. Data aggregate 2 independent experiments. **(H)** Huh7 cells were infected with HCV at MOI = 1 and cultured with or without 100 µM 6-AN ± 500 µM NAM and prepared for observation by transmission electron microscopy. Red arrows indicate DMVs.

cell remains largely unchanged, their size, and potentially their functional activity, is reduced when nicotinamide metabolism is inhibited. Such a reduction in replication complex size likely contributes to the overall decrease in viral replication observed under these conditions. HCV replicates in membranous structures characterized by double-membrane vesicles (DMVs) that are induced by viral replication. We thus specifically analysed the formation of DMVs by electron microscopy in Huh7 cells infected with HCV. Fig 1H shows representative pictures of infected cells 72 h post-infection, where the presence of DMV structures is indicated by the red arrows. While DMVs are visible in control condition, 6-AN treatment strongly inhibits the formation of these structures. Interestingly, the addition of NAM restored the formation of DMVs. Thus, the inhibition of NAD(H) metabolism by 6-AN prevents the formation of DMVs which are required for HCV replication.

## HCV replication depends on the NAD(H) salvage pathway

We then assessed the contribution of the different pathways involved in the synthesis of NAD(H) to HCV replication (Fig 2A). We first verified whether HCV infection modulated the expression of enzymes involved in this pathways. We quantified NAMPT, NAPRT, and QPRT by RT-qPCR in Huh7 cells after 3 days of infection. NAMPT is the rate-limiting enzyme of the nicotinamide salvage pathway, and NAPRT and QPRT are required for the Preiss-Handler and de novo NAD(H) synthesis pathways, respectively (Fig 1A). None of these enzymes was significantly modulated upon infection (S2A–S2C Fig). These results suggest that sensitivity to inhibitors of the nicotinamide metabolic pathway does not appear to be affected by viral infection. Furthermore, these observations are in agreement with transcriptomic data obtained from biopsies of patients infected or not by HCV [32] (GSE84346)), and showing no significant difference in the expression of these three enzymes, in HCV-infected patients compared to non-infected patients (S2D–S2F Fig). Altogether, these observations suggest that sensitivity to inhibitors of the nicotinamide metabolic pathway does not appear to be affected by viral infection. Because the salvage pathway depends on the reaction catalyzed by NAMPT (Fig 1A), we first evaluated a direct inhibitor of this enzyme, namely FK866. As shown in Fig 2A, FK866 suppressed viral replication and similar results were obtained with STF-118804, another inhibitor of NAMPT (Fig 1B). We quantified total NAD in cells treated with the antimetabolite 6-AN or the NAMPT inhibitor FK866. We used a bioluminescent assay measuring total NAD+ and NADH with high sensitivity through an enzymatic cycling system in which NAD+ is converted to NADH, enabling reductase-dependent generation of luciferin and subsequent luminescence proportional to total NAD cellular content. Intracellular NAD-dependent luminescence decreased by 24% in 6-AN–treated cells and by 97% in FK866-treated cells (S3 Fig). In both cases, NAM supplementation restored NAD levels. These results are consistent with the distinct mechanisms of action of the two compounds. FK-866 is a potent inhibitor of NAMPT, the rate-limiting enzyme of the nicotinamide salvage pathway, and therefore induces a profound depletion of intracellular NAD. In contrast, 6-AN acts as an antimetabolite that competes with nicotinamide and leads to the synthesis of 6-ANAD. Because it does not inhibit NAMPT directly, 6-AN results in only a modest reduction in NAD levels while still impairing NAD-dependent enzymes through 6-ANAD accumulation. Similarly to 6-AN, FK866 and STF-118804 induced a marked reduction in intracellular ATP levels and a concomitant inhibition of cell proliferation, reflecting a global impairment of cellular metabolism. Both effects were fully reversed by

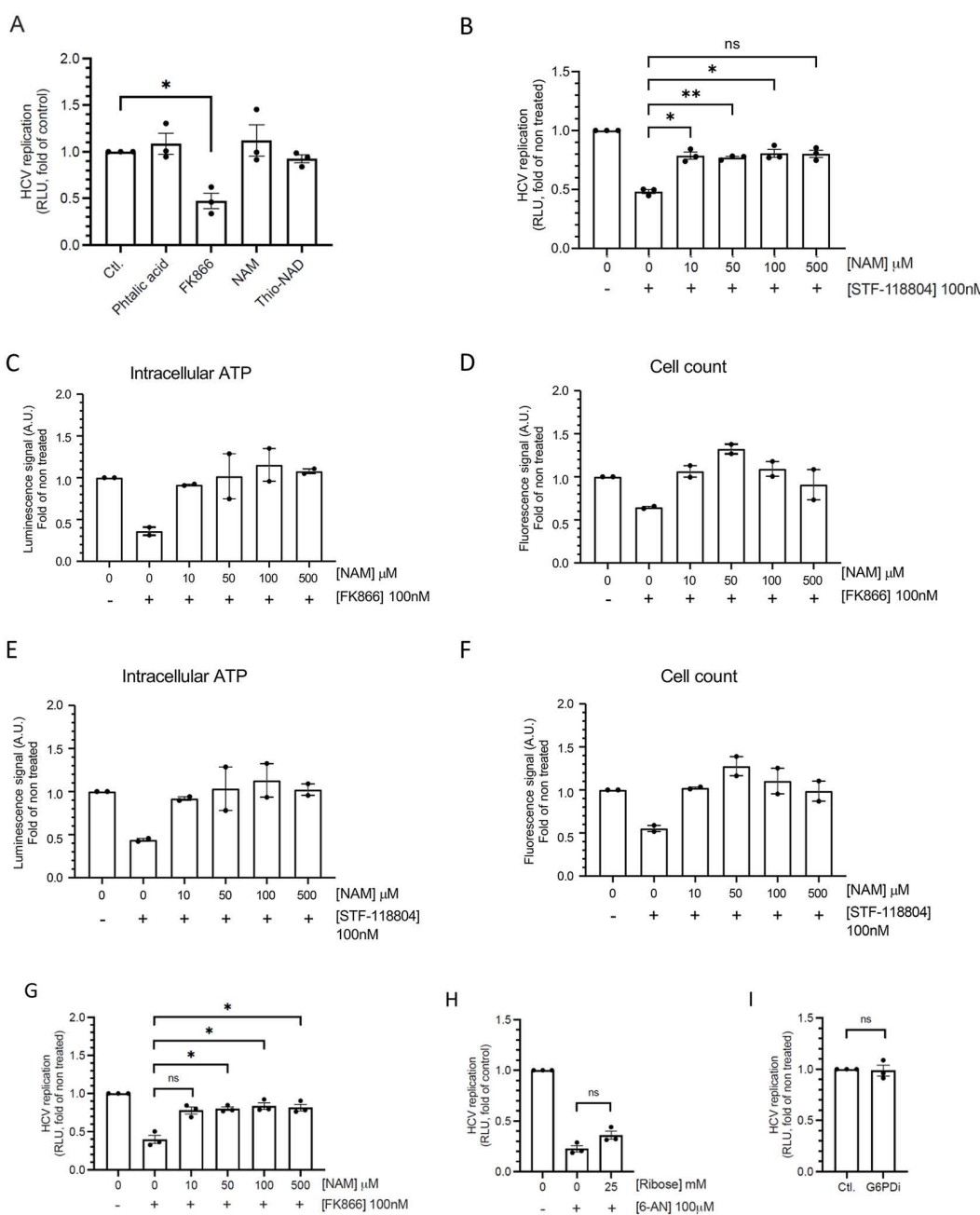

**Fig 2. HCV replication depends on nicotinamide salvage pathway. (A-B)** Huh7 cells were electroporated with subgenomic replicon and cultured for 72 h with the indicated treatment. Renilla luciferase activity was measured in cell homogenates and viral replication normalized to non-treated control (0 or Ctl.). Are presented means ± SEM. **(A)** Cells were cultured with or without 100 µM of thio-NAD, 100 µM of phthalic acid, 100 nM of FK866 or 500 µM of NAM, n = 3, one sample t-test, Bonferroni- Sidák adjusted p-value for multiple comparison to Ctl. **(B)** Cells were cultured with or without 100 nM STF-118804 and increasing doses of NAM. n = 3, one-way ANOVA, Bonferroni-Sidák adjusted p-value for multiple comparison to condition with STF-118804 alone. **(C-F)** Huh7 cells were cultured with or without 100 nM of FK866 or STF-118804 and increasing doses of NAM (0, 10, 50, 100 or 500 µM). Intracellular ATP amounts **(C and E)** were determined using CellTiter Glo assay (Promega) and cell proliferation **(D and F)** after Hoechst staining of nuclei and quantification of fluorescence. Are presented means ± SEM, n = 2. **(G)** Cells were electroporated with subgenomic replicon and cultured for 72 h with or without 100 nM of FK866 and increasing doses of NAM, n = 3, one sample t-test, Bonferroni-Sidák adjusted p-value for multiple comparison to condition with FK866 alone. **(H)** Cells were cultured with or without 100 µM 6-AN ± 25 mM of Ribose, n = 3, one-way ANOVA, Bonferroni adjusted p-value for multiple comparison. **(I)** Cells were cultured with or without 100 nM of G6PDi, n = 3, one sample t-test. n.s. non significative, *p < 0.05, **p < 0.01.

supplementation of the culture medium with NAM (Fig 2C–2F). Importantly, the antiviral activity of FK866 and STF-118804 was also abrogated by the addition of NAM, demonstrating that inhibition of NAMPT is responsible for this phenotype. The resulting metabolic slowdown creates a cellular environment that is probably less permissive to viral replication. We also sought to determine whether inhibition of nicotinamide metabolism could have an effect on infection once it was established, and not only during the initial phases of infection. To do this, we infected the cells at a MOI of 1, let the infection develop for 72 hours before treating the cultures with 6-AN or FK866 supplemented or not with NAM. Although an effect of 6-AN on intracellular RNA levels was observed, no effect of FK-866 was detected (S4A Fig). Furthermore, we observed no effect of the inhibitors on either the quantity or the quality of secreted viral particles (S4B and S4C Fig). These results indicate that nicotinamide metabolism plays a critical role during the establishment phase of HCV infection, while its contribution is limited once the infection is fully established. To determine whether drugs targeting nicotinamide metabolism could alter viral replication in quiescent cells, Huh7 cells were differentiated for 7 days in the presence of 2% DMSO, and were then infected with HCV as previously described ([34]; S5 Fig). In this model, treatment with 6-AN or FK866 reduced intracellular levels of viral RNA (S5A Fig). Furthermore, and despite a limited effect on viral RNA secretion, the production of infectious particles collapsed (S5B and S5C Fig). This inhibitory effect was reversed by adding NAM to the culture medium, confirming the role of nicotinamide metabolism in HCV replication. We have also tested the effect of inhibition of NAD(H) metabolism in HepaSH cells, which are human hepatocytes purified from the liver of immunocompromised mice transplanted with primary human hepatocytes (PHH) [33]. Freshly purified cells were seeded in glycogen-coated well plates and cultured for 4 days prior to infection at a MOI of 1. Three days after infection, intracellular and secreted HCV RNA levels were quantified (S6 Fig); although viral titers remained below the detection threshold, quantifiable levels of extracellular viral RNA were detected. Under these conditions, HCV infection was reduced upon 6-AN treatment, with a more pronounced effect observed on intracellular viral RNA than on secreted RNA. Supplementation with NAM restored extracellular viral RNA levels, while only partially recovering intracellular genome levels. Taken together, and despite the overall low level of infection in this experimental setting, these data support an inhibitory effect of 6-AN on HCV replication in PHH. As NAD(H) can also be synthesized from tryptophan via the *de novo* biosynthesis pathway (Fig 1A), phthalic acid was used to inhibit Quinolinic acid PhosphoRibosylTransferase (QPRT), an enzyme catalyzing the last reaction leading to the de novo synthesis of nicotinic acid mononucleotide (NaMN) [34,35]. This drug had no effect on HCV replication in Huh7 cells (Fig 2A), suggesting that the *de novo* biosynthesis pathway is not required for viral replication. We also showed that viral replication is not affected by thionicotinamide adenine dinucleotide (Thio-NAD; S1D Fig), an analogue of NAD(H) inhibiting NADPK, the enzyme converting NAD to NADP [36](Fig 2A). Neither phthalic acid nor Thio-NAD had an effect on intracellular ATP or cell proliferation (S7 Fig). This suggests that conversion of NAD(H) into NADP(H) is not critical in the antiviral effect of 6-AN. 6-AN is also used as an inhibitor of the pentose phosphate (PPP) pathway via the inhibition of glucose-6-phosphate dehydrogenase (G6PD) and 6-phosphogluconate dehydrogenase (6PGD). D-ribose was thus added to the culture medium to determine whether inhibition of the PPP pathway contributes to the restriction of HCV replication by 6-AN. Indeed, phosphorylation of D-ribose by the ribokinase RBKS produces R5P, which enables nucleotide biosynthesis when the PPP pathway is impaired. HCV replication was weakly restored by D-ribose, suggesting that the effect of 6-AN is essentially independent of PPP pathway inhibition (Fig 2H) [14,37]. Furthermore, HY-W107464, a specific inhibitor of G6PD did not inhibit HCV replication (Fig 2I), confirming that PPP inhibition was not involved in HCV inhibition by 6-AN. Overall, these observations indicate that HCV critically depends on a functional NAD(H) salvage pathway for its replication and that NAMPT activity, the limiting enzyme of this pathway, is essential for HCV replication.

## Inhibition of the NAD(H) pathway reduces the secretion and specific infectivity of infectious particles

As the inhibition of NAD(H) biosynthesis affects HCV replication as assessed with subgenomic replicon, we wanted to evaluate the impact on the production of new virions. Thus, Huh7 cells were infected with full-length HCV virions and treated cells with 6-AN in the absence or presence of NAM. For technical reasons, we used the strain Jc1-E2Flag [25] at

a MOI 1. After 72 h of culture, virus titration showed that 6-AN decreased by 2 logs in the production of infectious particles and this inhibition was reversed by NAM (Fig 3A). The same observations were obtained with the NAMPT inhibitor FK866 (Fig 3D). Surprisingly, in presence of 6-AN, the amount of viral RNA secreted did not decrease as much as the number of infectious particles produced (Fig 3B), suggesting a change in particle infectivity. Indeed, the ratio between infectious particles and secreted viral genomes (FFU/GE), shown in Fig 3C and 3F was markedly reduced for cells treated with 6-AN or FK866 respectively. As the FFU/GE ratio reflects the specific infectivity of the produced particles, this decrease suggests a reduced infective capacity of the virions, or at least a reduction in the number of infectious particles secreted, consistent with the inhibitory effect initially observed on viral replication (Fig 1B–1D) and DMV formation (Fig 1H). The addition of NAM reversed this phenotype as expected (Fig 3C). The same results were obtained with FK866 (Fig 3E and 3F). A comparable inhibitory effect was also observed in Huh7 cells differentiated with DMSO prior to infection (S5 Fig). Thus, inhibition of the NAD(H) salvage pathway reduced not only the overall production of infectious particles but also changed their capacity to propagate infection (Fig 3). In contrast and as expected, Thio-NAD and Phtalic acid had no effect on the production of infectious HCV particles (Fig 3G), confirming the results obtained with the subgenomic replicon (Fig 2A).

**6-aminonicotinamide inhibits central carbon metabolism**

We then determined the effect of 6-AN on the metabolism of Huh7 cells to better understand how inhibition of NAD(H) biosynthesis reduces HCV replication and the production of infectious particles. We first analyzed polar and semi-polar metabolites in Huh7 cells treated or not with 6-AN at 100 µM. Of the 1,768 metabolites detected by LC-MS/MS, 151 were identified with a high level of confidence (annotation level 1 and 2a) (S1 Table). Principal component analysis of experimental replicates based on these 151 metabolites revealed a clear segregation between untreated and 6-AN-treated cells (S8A Fig). The PC1 variation between controls and 6-AN-treated cells suggests that the main differences between the two clusters are captured by a unique set of metabolites explaining the effect. Sixty-eight metabolites were significantly downregulated and 19 were upregulated (Fig 4A; fold change>20% and p-value<0.05). First, as expected, the levels of ATP, NAD$^+$ and NAM were reduced after 6-AN treatment. Interestingly, the first metabolites of glycolysis (glucose and glucose-6-phosphate) were increased, while final products such as pyruvate and lactate were decreased (Fig 4B). This may reflect a downregulation in glucose consumption rate and an overall reduced metabolic activity, contributing to the drop in ATP levels observed in the presence of 6-AN. Moreover, the amounts of several metabolites of the TCA cycle were reduced, suggesting that mitochondrial metabolic activity is also affected (Fig 4C).

To confirm the effect of 6-AN on glycolysis and mitochondrial respiration, both activities were measured after a 72 h of treatment, using an extracellular flux analyzer [38]. In cell cultures, glycolysis and mitochondria-derived $CO_2$ are the two main contributors to extracellular acidification rate (ECAR). We therefore determined first the glycolytic proton efflux rate (glycoPER) to monitor the glycolytic activity of cells [39]. Basal glycolytic activity was measured first, followed by compensatory glycolysis after blockade of complex I and III of the mitochondrial respiratory chain with Rotenone and Antimycin A, respectively. In agreement with LC-MS/MS data, 6-AN treatment induced a strong inhibition of both basal and compensatory glycolysis, which are restored by NAM (Fig 5A and 5B). We then analyzed the oxygen consumption rate (OCR) to determine the effect of 6-AN treatment on mitochondrial respiration (Fig 5C and 5D). In cells treated with 6-AN, basal respiration increased, while the addition of NAM reversed the effect of 6-AN (Fig 5C and 5D). Using sequential addition of inhibitors, we determined as previously described the fraction of $O_2$ consumption utilized for ATP production, the maximal respiration rate and the spare respiration capacity (Fig 5E). We observed that in presence of 6-AN, the increase in basal respiration correlates with an increase in ATP production through oxidative phosphorylation (Fig 5D and 5E). Thus, the increase in oxidative phosphorylation aims to compensate the glycolytic blockade. We also observed a total loss of the spare respiratory capacity in 6-AN-treated cells (Fig 5F). This mitochondrial parameter reflects the cell's ability to respond to enhanced energy demand under a physiological stress by stimulating the respiratory chain. In presence of 6-AN, oxidative phosphorylation is increased to its maximum, matching levels reached by addition of the uncoupling agent FCCP.

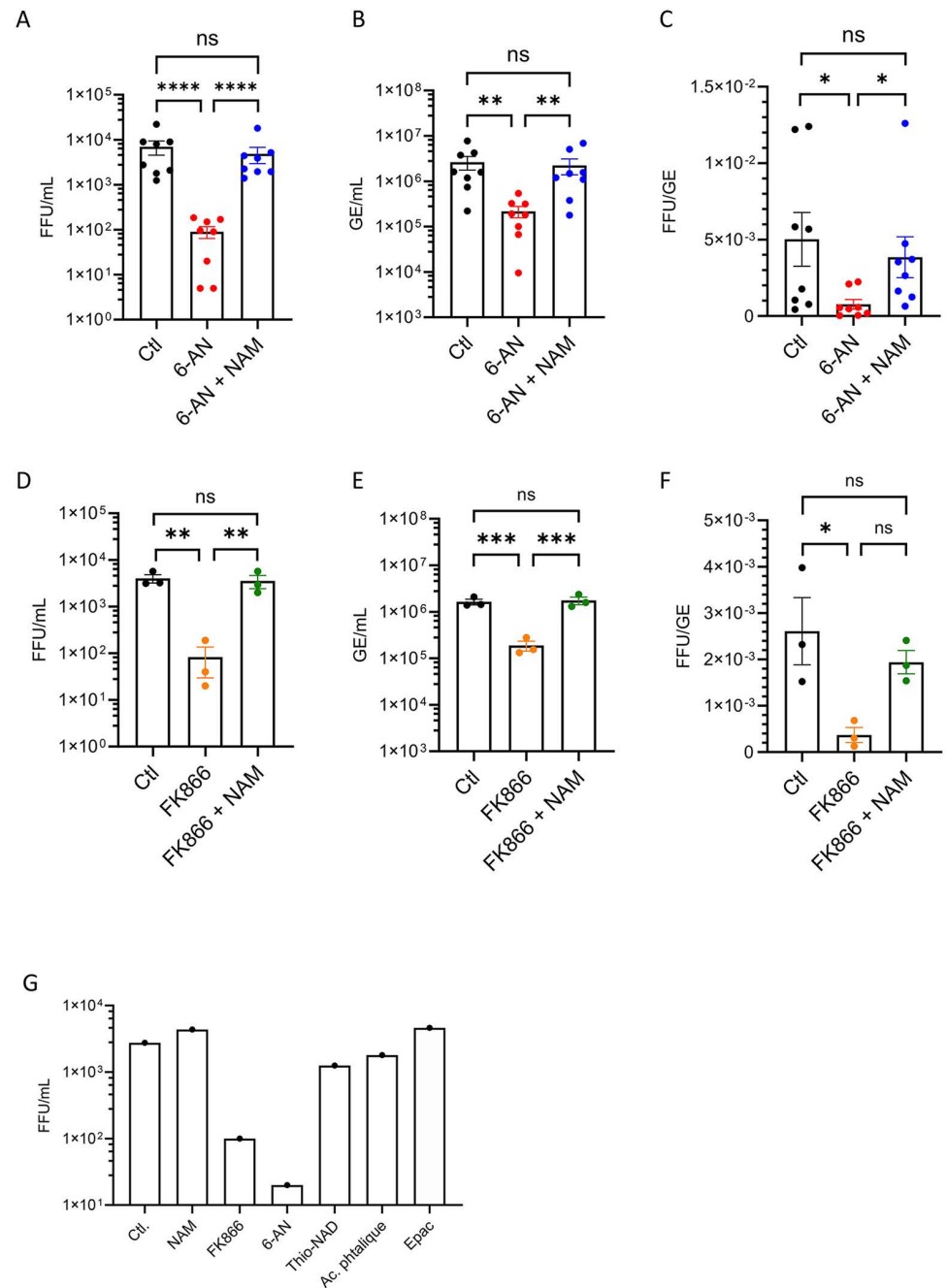

**Fig 3. Inhibition of NAD(H) metabolism reduces the production of infectious HCV particles. (A-C)** Huh7 cells were infected with HCV Jc1 strain at MOI 1 and cultured 72 h with or without 6-AN (100 μM), in the absence or presence of NAM (500 μM). **(A)** FFU were determined from cell culture supernatant harvested at 72 h post-infection. **(B)** Viral genomes within culture supernatants at 72 h post-infection. **(C)** Specific infectivity of secreted viral particles. Are presented means ± SEM, n = 8, one-way ANOVA for multiple comparison, n.s. non significative, *p < 0.05, **p < 0.01, ****p < 0.0001. **(D-F)** Huh7 cells were infected with HCV at MOI = 1 and cultured for 72 h with or without 100 nM FK866 or 100 nM FK866 + 500 μM NAM. **(D)** FFU were determined from cell culture supernatant harvested at 72 h post-infection. **(E)** Viral genomes within cell culture supernatants at 72 h post-infection. **(F)** Specific infectivity of viral particles. **(D-F)** Are presented means ± SEM, n = 3, one-way ANOVA for multiple comparison, n.s. non significative, *p < 0.05, **p < 0.01, ***p < 0.001. **(G)** Huh7 cells were infected with HCV at MOI = 1 and cultured with or without 100μM 6-AN, 100 μM thio-NAD, 100 μM phthalic acid, 100 nM FK866, or 500 μM NAM. Infectious titer was determined in cell culture supernatants at 72 h post-infection.

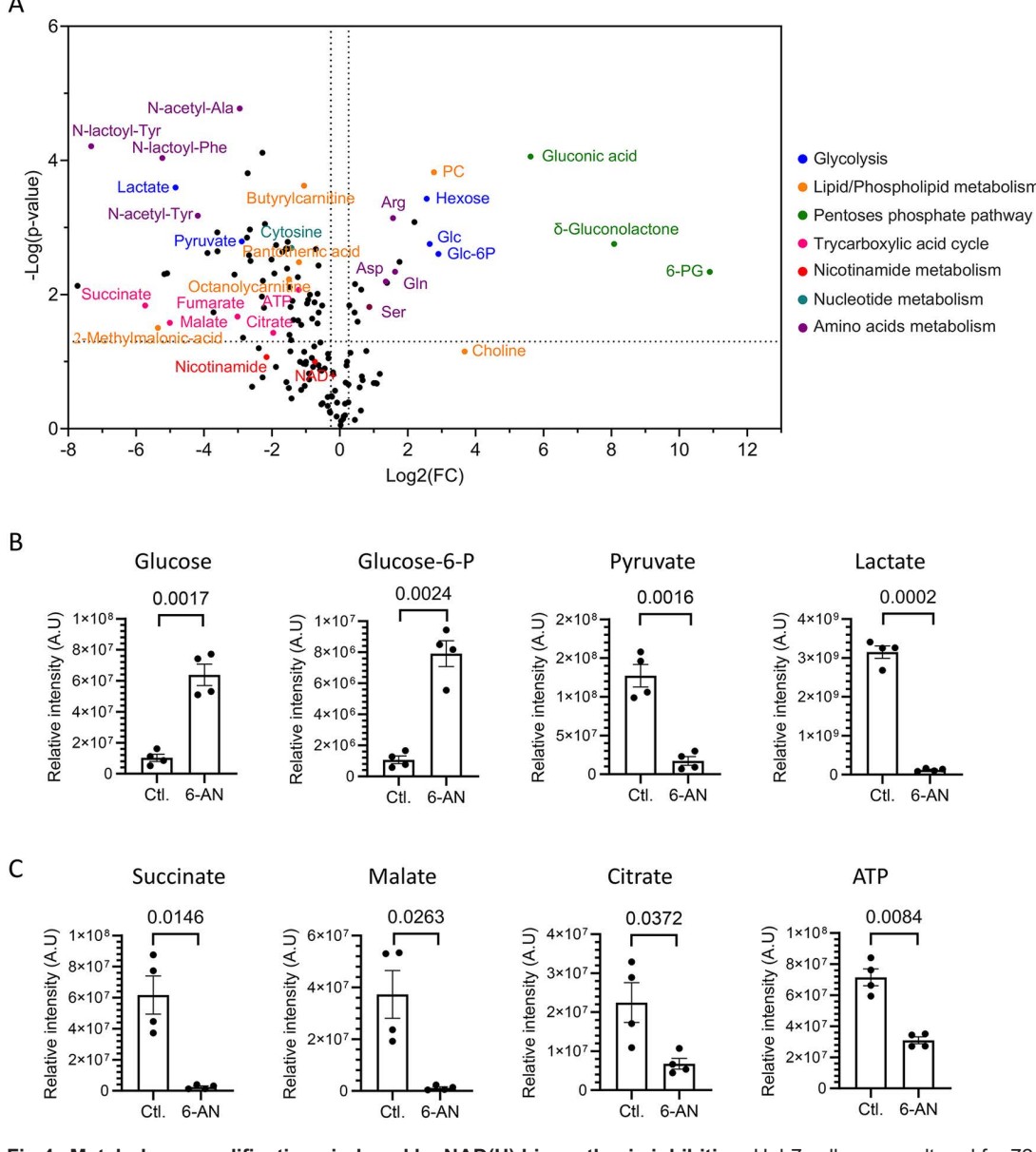

**Fig 4. Metabolome modifications induced by NAD(H) biosynthesis inhibition.** Huh7 cells were cultured for 72 h with or without 6-AN (100 µM) before metabolite extraction. **(A)** Volcano plot showing variations in the expression of the 151 metabolites identified with high confidence criteria. The x-axis corresponds to $Log_2$(fold change) and the y-axis corresponds to -$Log_{10}$($p$-value). Horizontal dotted line indicates the $p$-value threshold of 0.05 and vertical dotted lines indicate variations above the 20% threshold. Metabolites were grouped by color for their association with a given metabolic pathway. **(B and C)** Relative abundances of metabolites associated to glycolysis and TCA cycle. Means±SEM obtained from 4 independent experiments are indicated with $p$-value for Student's t-test comparison.

Overall, these results show reduced glycolytic activity, which could limit viral replication, combined with increased mitochondrial respiration as compensatory mechanism for ATP production.

### Inhibition of NAD(H) biosynthesis reduces cellular triglycerides level and secretion of apolipoproteins

LC-MS/MS data also showed modulation of several lipid metabolites (Fig 4A), probably a consequence of the overall effect of 6-AN on the cellular energetic metabolism. In particular, a metabolite set enrichment analysis (MSEA) showed

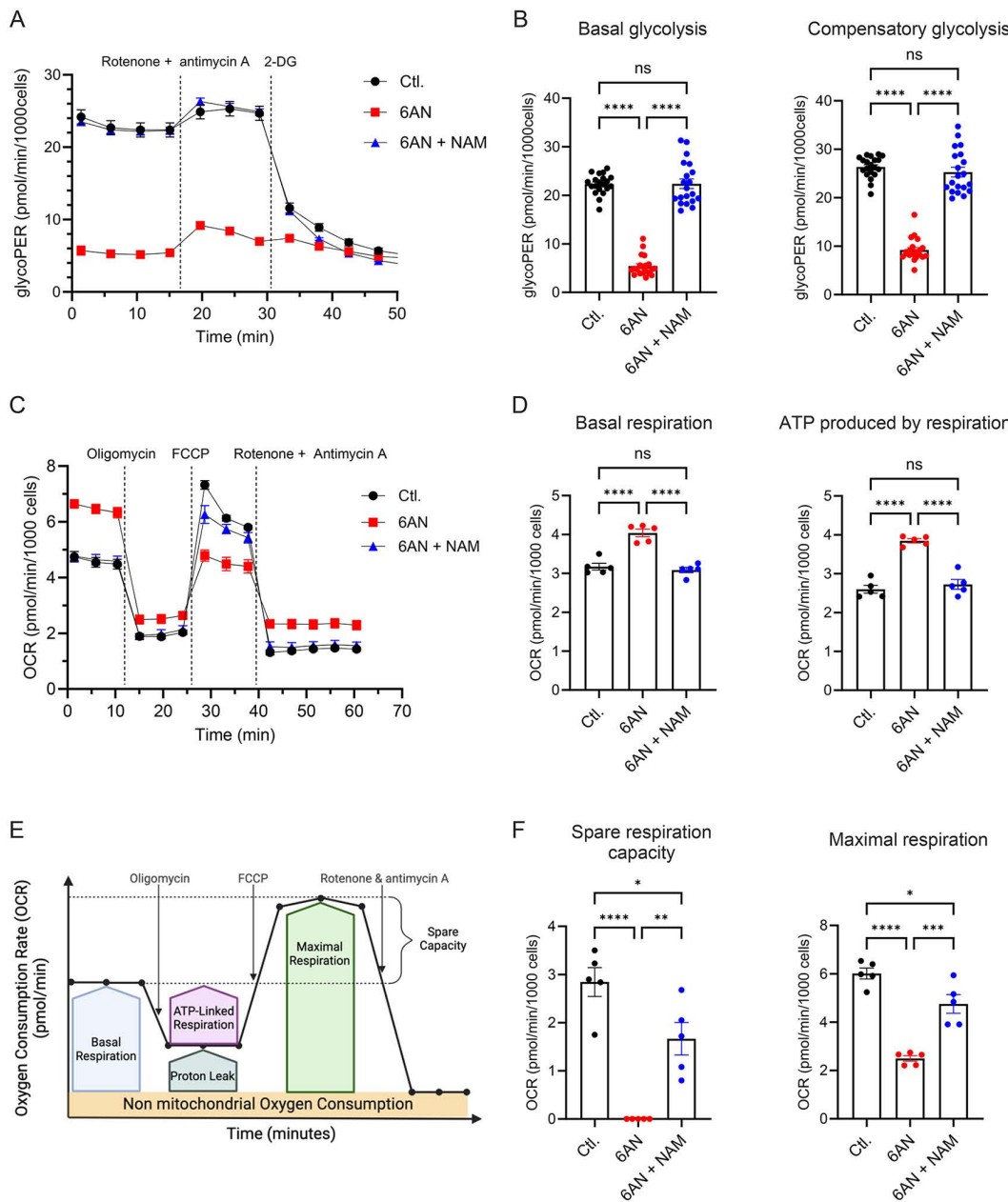

**Fig 5. 6-AN inhibits glycolysis and rebalances mitochondrial respiration.** Huh7 cells were treated or not with 100 μM 6-AN±500 μM of NAM for 72 h. **(A)** Quantification of glycolytic activity determined by the GlycoPER (proton efflux rate specific to glycolysis) using the Glycolytic Rate Test. Are presented means±SEM of 20 replicates. Data are representative of 3 independent experiments. **(B)** Basal glycolysis and compensatory glycolysis were calculated from the Glycolytic Rate Test presented in A (one-way ANOVA for multiple comparison, n.s. non significative, ****$p < 0.0001$). **(C)** Oxygen consumption rate (OCR) was determined using the Mito Stress test before and after addition of oligomycin (Complex V inhibitor), FCCP (uncoupling agent), rotenone (Rot; Complex I inhibitor) plus antimycin A (Anti-A; Complex III inhibitor). **(D)** Basal OCR and ATP production were calculated from **(C)**, as indicated in E. **(E)** Graphical representation of the different parameters that can be calculated with the Mito Stress assay from Agilent. **(F)** Spare respiration capacity and maximal respiration were calculated from C. **(C, D and F)** Are presented means±SEM of 5 replicates (one-way ANOVA for multiple comparison, *$p < 0.05$, **$p < 0.01$, ***$p < 0.0002$, ****$p < 0.0001$). Data are representative of 3 independent experiments.

that phosphatidylcholine and phosphatidylethanolamine biosynthesis pathways were strongly impacted by 6-AN (S8B Fig). Levels of carnitine-conjugated fatty acids were also reduced, suggesting changes in fatty acid and triglyceride metabolism (Figs 4A and S8B). Given their key role in the formation of LVPs, we investigated the effect of 6-AN on apolipoproteins secretion and intracellular levels of triglycerides. After 72 h of 6-AN treatment, the secretion of ApoB and ApoE, which are associated with LVPs and are essential to their infectivity, was reduced (Fig 6A and 6B). We also observed a reduction in intracellular TG content (Fig 6C). We thus studied the impact of 6-AN treatment on intracellular lipid droplets by confocal microscopy after staining of neutral lipids. The number and morphological parameters of lipid droplets (LDs) were determined after three-dimensional reconstruction from fluorescence microscopy images (Fig 6D–6E). Interestingly, the average LD volume per cell was reduced by 62% in 6-AN–treated cells (Fig 6E), consistent with intracellular TG quantification using a specific biochemical enzymatic assay (Fig 6A). Addition of NAM to the cell culture medium barely restored the volume of the LDs (Fig 6E). Altogether, these data indicate an important impact of NAD(H) metabolism inhibition on hepatocyte neutral lipids metabolism decreasing intracellular TG content and lipoprotein secretion, with a potential impact on the amount and quality of secreted of LVPs.

### Inhibition of NAD(H) biosynthesis alters the composition and buoyant density of HCV secreted particles

Based on the effect of 6-AN on lipid metabolism, we hypothesized that 6-AN could affect LVP production. We therefore infected cells with the Jc1-E2Flag strain, treated them with or without 6-AN, and then separated secreted HCV particles on a density gradient. We then collected fractions of increasing density, and analyzed the specific infectivity of HCV particles in each fraction. In low density fractions (1.00-1.06 g/ml), specific infectivity was reduced by up to 90% by 6-AN, whereas it was unaffected in higher-density fractions (Fig 7A). This effect on low-density viral particles was reversed by NAM. As previously observed in whole culture supernatants, the overall amount of viral RNA secreted and FFU titer were decreased in the presence of 6-AN, regardless the density of the viral particles (Fig 7B and 7C). These results were confirmed when we analyzed the specific infectivity of LVPs produced by infected cells with the Jc1 (S9 Fig). Therefore, in addition to inhibiting viral replication, treatment with 6-AN specifically reduced the production of LVPs.

## Discussion

An increasing body of literature is focusing on the mechanisms by which viruses manipulate cellular metabolism to foster their replication. HCV is a remarkable example of an hepatotropic virus subverting cellular metabolism as it deeply interferes with glucose and lipid metabolism, and hijacks lipoprotein secretion pathways to form LVPs. Previous studies have shown that infection of hepatocytes by viruses can induce glycolysis [1,6,7]. In particular, we have established that DENV infection triggers a glycolytic response in hepatocytes, and that targeting the NAD(H) biosynthesis pathway with drugs reduces viral replication [8]. Similarly, ZIKV infection of brain glial cells is inhibited by blocking the NAD(H) metabolic pathway [14]. Since HCV also belongs to the *Flaviviridae* family, alongside DENV and ZIKV, we investigated whether HCV replication similarly depends on NAD(H) biosynthesis. Our results demonstrate that HCV is indeed dependent on the NAD(H) biosynthesis salvage pathway for its replication and for the secretion of highly infectious LVPs. Inhibition of this pathway by 6-AN inhibits glycolysis and rewires central carbon metabolism resulting in a reduction of intracellular TG storage and decreased of ApoB and E, both of which are essential for the infectivity of LVPs (Fig 8).

Viruses critically rely on central carbon metabolism for their replication. Many enzymes involved in these pathways use NAD(H) as a cofactor to transfer electrons during redox reactions. For instance, in glycolysis, glyceraldehyde-3-phosphate is oxidized to 1,3-bisphosphoglycerate, while $NAD^+$ is reduced to NADH. Several steps in the TCA cycle, such as isocitrate conversion to α-ketoglutarate, also contribute to NADH production. This $NAD^+$/NADH cycling plays a central role in cellular respiration since NADH is oxidized by complex I providing electrons to the mitochondrial respiratory chain, that ultimately drives oxidative phosphorylation and ATP production. We observed a strong decrease in the glycolytic activity of the cells treated with 6-AN (Fig 5). As a result, glycolysis can no longer contribute to meeting the cell's ATP requirements.

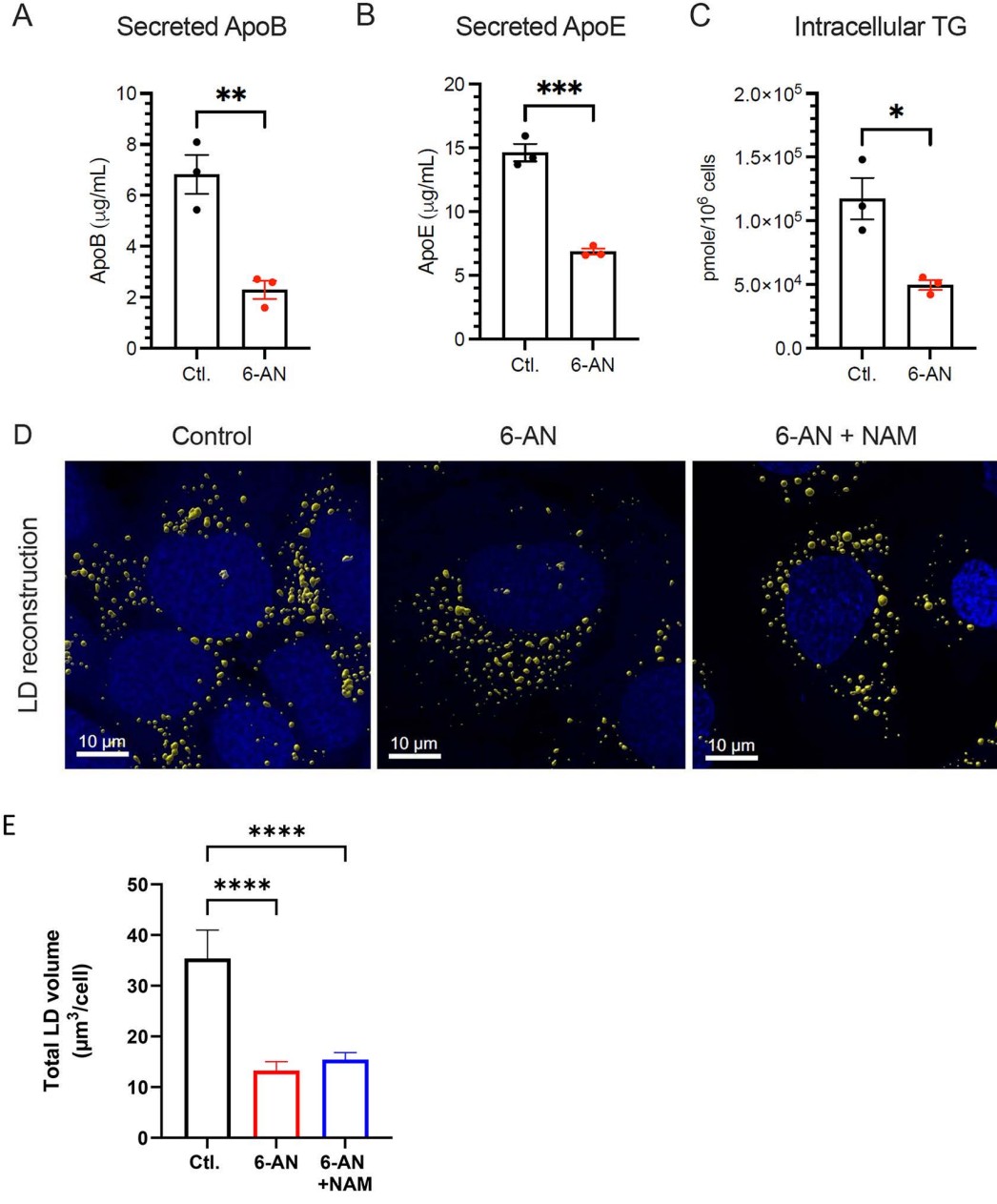

**Fig 6. Inhibition of NAD(H) metabolism decreases intracellular TG storage.** Huh7 cells were treated or not with 100 μM 6-AN for 72h of culture. **(A-B)** ApoB and ApoE secreted in cells supernatant quantified by ELISA. Are presented means±SEM (n=3, student's t-test, **p<0.0021, ***p<0.0002). **(C)** Quantification of intracellular triglycerides in total cell extracts. Are presented means±SEM (n=3, student's t-test, *p<0.01). **(D)** Representative picture of 3D lipid droplets reconstruction using IMARIS software in Huh7 cells after Oil-red-O (yellow) and Hoechst (blue) staining. **(E)** The mean total volume of lipid droplets per cell was determined by analyzing their intracellular content of 50 cells for each condition with the IMARIS software. Are presented means±SEM, one-way ANOVA for multiple comparison, n.s. non significative, ****p<0.0001.

As a compensatory mechanism, mitochondrial respiration increases, supporting ATP production for the cell's metabolic activity (Fig 5). However, ATP level remains lower in 6-AN-treated cells (Fig 4A) as maximal respiration capacity is reached. In addition, the pool of NAD(H) is reduced in the presence of 6-AN, which also limits mitochondrial functions. Indeed,

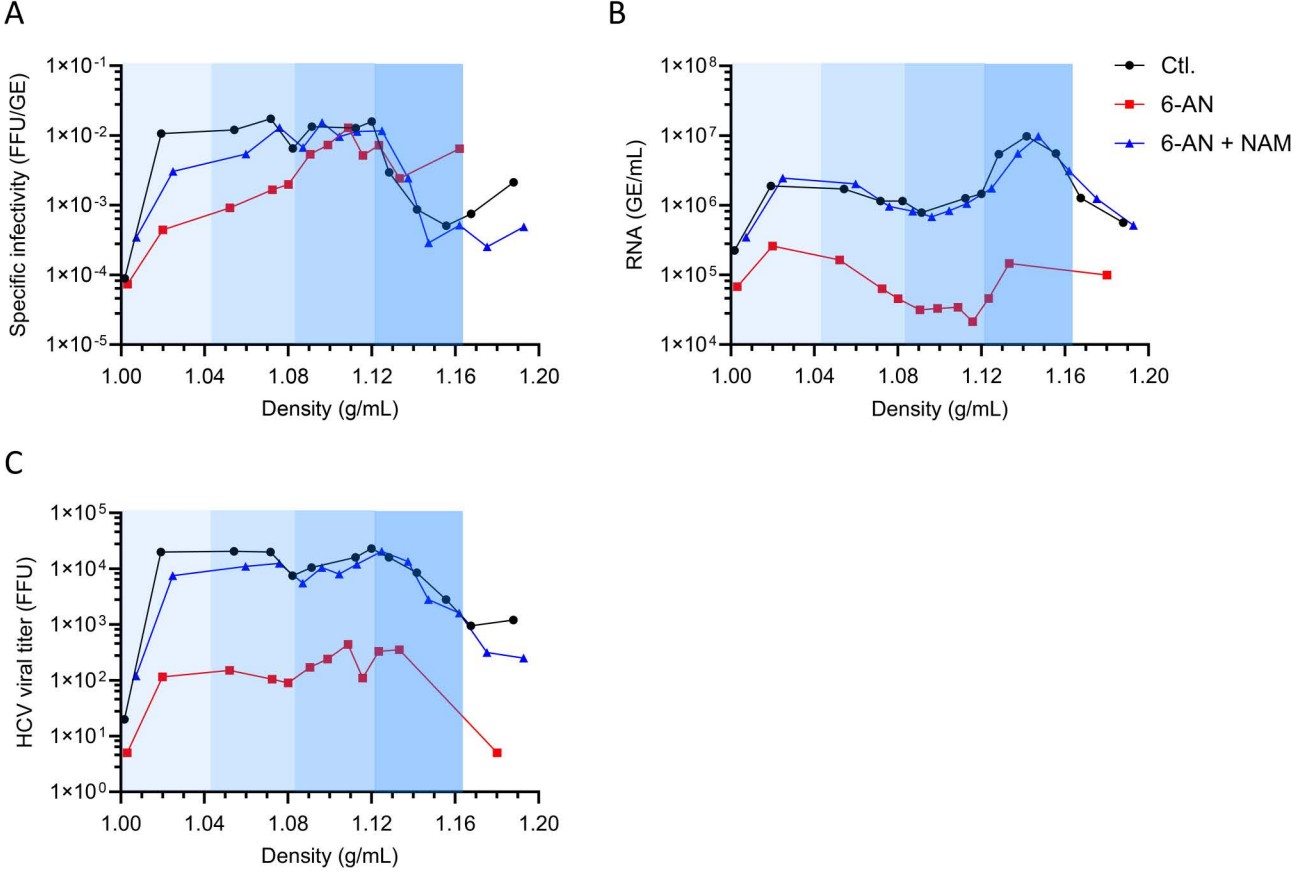

**Fig 7. Inhibition of NAD(H) metabolism reduces LVP infectivity. (A-C)** Huh7 cells were infected with HCV at MOI = 1 and cultured for 72 h with or without 6-AN (100 μM), in the absence or presence of NAM (500 μM). 1ml of ten-fold concentrated cell-culture supernatants were separated on iodixanol gradient (3-40%). Specific infectivity of viral particles **(A)**, HCV RNA content **(B)** and FFU **(C)** were determined in each of the collected density fractions. Are presented data representative of 3 independent experiments.

metabolomic data showed that several TCA metabolites were markedly reduced, likely reflecting their consumption and depletion. This reduction may impair the activity of succinate dehydrogenase—an enzyme that links the TCA cycle to the respiratory chain—thereby limiting electron transfer to complex II during the conversion of succinate to fumarate. Thus, by reducing NAD(H) availability, 6-AN not only inhibits glycolysis — and therefore reduces the energy derived from glucose — but also limits the cell's ability to increase mitochondrial respiration as a compensatory mechanism. Thus, decreasing the availability of NAD(H) broadly alters cellular metabolism, thereby reducing the capacity of cells to perform specific metabolic functions. Although this inhibition limits glycolysis and TCA cycle activity (Fig 5), leading to decreased cell proliferation, it does not compromise cell viability or cellular integrity (S1 Fig). Inhibition of the NAD(H) metabolism also inhibits HCV replication, and this effect is particularly pronounced when considering the initial phase of culture infection in contrast to latter stages when infection is established within the culture (S4 Fig). Inhibitory effect on viral life cycle is also specifically important on the production of infectious particles and the specific infectivity of secreted virions (Fig 3). This suggests that multiple steps necessary to HCV replication cycle are impaired, leading to cumulative effects on the production of viral particles. We have also shown that a similar effect is observed in DMSO-differentiated cells (S5 Fig). This metabolic constraint could therefore potentially impair both the synthesis of membranes required for the formation of double-membrane vesicles (DMVs) and the production of triglycerides essential for the biogenesis of lipo-viro-particles (LVPs).

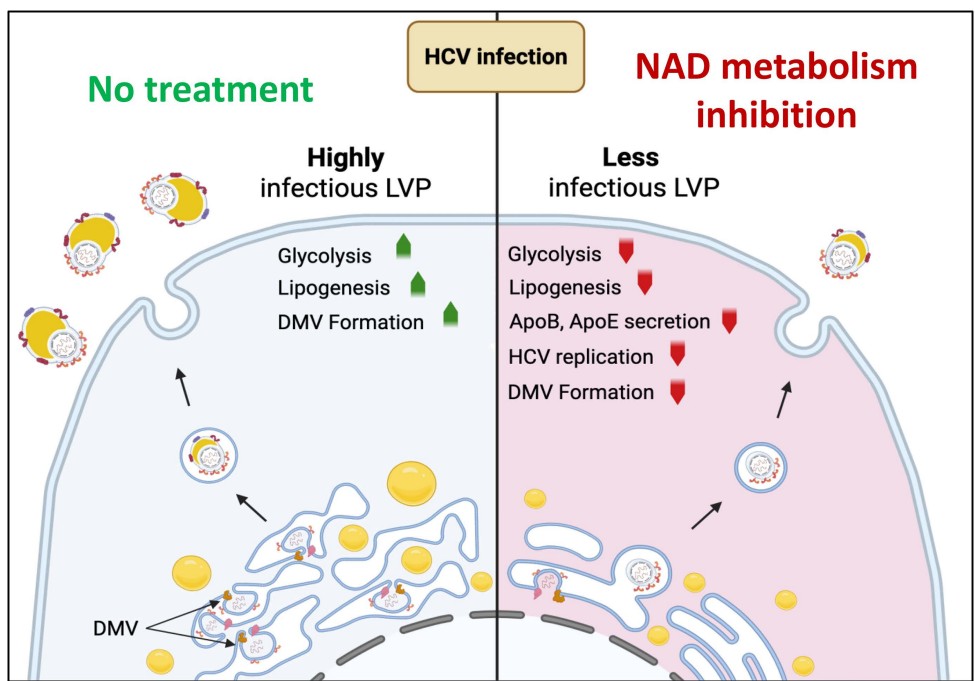

**Fig 8. Impact of NAD metabolism inhibition on HCV infection.** In the absence of treatment, HCV infection induces the lipogenesis, DMV formation and secretion of highly infectious Lipo-Viral-Particles (left side). In the presence of NAD metabolism inhibitors, glycolysis, lipogenesis, and apolipoprotein secretion metabolism are reduced, affecting the infectivity of the Lipo-Viral-Particles produced (right side). Image created in BioRender. Vidalain, P. (2026) https://BioRender.com/vv32jiw.

*Flaviviridae*, as well as other RNA viruses such as SARS-CoV-2, remodel endoplasmic reticulum (ER) membranes into double-membrane vesicles (DMVs), which serve as viral replication organelles. These DMVs concentrate viral components necessary for genome replication and particle maturation while protecting them from innate immune detection [40]. HCV-induced DMVs are well established as the primary sites of viral RNA replication and are closely associated with downstream steps of the viral life cycle, although the precise spatial organization and coupling of replication, assembly, and lipidic envelopment remain incompletely understood. Viral RNA replication occurs within DMVs enriched in the nonstructural proteins NS3, NS4B, NS5A, and NS5B, whereas virion assembly is thought to take place at endoplasmic reticulum (ER) membranes in close proximity to lipid droplets (LDs), involving the structural proteins core, E1, and E2 [17]. High-resolution live-cell imaging has revealed a selective recruitment of ER membranes around LDs, where structural proteins and components of the replication machinery colocalize [41]. These ER membranes are decorated with DMVs, providing a potential topological framework to facilitate the spatial coordination of HCV genome replication and virion assembly. Nevertheless, how DMVs, assembly sites, and LDs are functionally coordinated remains unclear, particularly regarding the transport of viral components between compartments and the involvement of host lipid transfer enzymes [17]. Notably, sharp differences in lipid composition have been identified between DMVs and the ER membranes from which they derive. For instance, HCV-induced DMVs contain more cholesterol than ER membranes [42]. Additionally, DMVs are enriched in specific phospholipids such as phosphatidylinositol-4-phosphate (PI4P) [43], phosphatidylcholine [44], and sphingomyelin [45]. A lipidomic analysis of cells infected with human coronavirus 229E (HCoV-229E) revealed widespread remodeling of intracellular lipid composition, including elevated levels of fatty acids and glycerophospholipids including phosphatidylcholine and phosphatidylserine [46]. It was also shown that

phospholipase A2α — an enzyme that generates lysophospholipids by cleaving the fatty acid in sn-2 position of glyc-erophospholipids — is relocalized to DMVs in HCoV-229E-infected Huh-7 cells, and that its activity is essential for viral replication [47]. In the context of HCV, it was found that infected cells are enriched in glycerophospholipids containing long-chain fatty acids [48]. Our metabolomic analysis showed that 6-AN also interfere with the transfer of acetyl groups into mitochondria (cf. S8 Fig). Considering the pivotal role of this transfer in the biosynthesis of glycerophospholipids, this suggests alterations in fatty acids synthesis.

The metabolite set enrichment analysis indicates that treatment with 6-AN interferes with the biosynthesis of glyc-erophospholipids such as phosphatidylcholine and phosphatidylserine (S8B Fig), both precursors of phosphatidic acid (PA). Moreover, we observed that phosphocholine (PC) and choline, two precursors of phosphatidylcholine, accumu-late in Huh7 cells treated with 6-AN (Fig 4A). This argues for an effect of 6-AN on phospholipids that are essential to the formation of HCV viral factories. Indeed, it has been demonstrated that PA is a critical lipid required for DMV formation in HCV-infected Huh7 cells [49]. PA can be generated either from glycerophospholipids degradation via phospholipase D (PLD) activity or by the transfer of one fatty acid to lysophosphatidate through the action of acylglycerol-3-phosphate acyltransferase (AGPAT) during glycerophospholipids and TG biosynthesis. PA plays a prominent role in membrane rearrangements mainly because of its unique biophysical properties: a small headgroup; negative charge; and a phosphomonoester group. PA can influence membrane fusion and fission by generating nega-tive membrane curvature [50]. Overall, this suggests that 6-AN may impair DMV formation by limiting the availability of essential lipid substrates. Given that the lipid composition of DMVs appears to be finely regulated [51], and that HCV has been shown to induce an increase in polyunsaturated fatty acids (PUFAs), such as arachidonic acid, docosahex-aenoic acid, and eicosapentaenoic acid, as well as to modulate the elongation and desaturation activities of fatty acids incorporated into membrane phospholipids [48], it would be of interest to perform a detailed lipidomic analysis of cells treated with 6-AN. Specifically, it would be valuable to determine which molecular species within different phospholipid classes are selectively modulated under conditions of NAD(H) metabolism inhibition. Likewise, analyzing the expres-sion changes of enzymes involved in these biosynthetic pathways under these conditions would be informative. Such analyses should help to better define the broader impact of lipid metabolism modulation on DMV formation during viral infections. Moreover, ApoE incorporation within LVP that is crucial for the infectivity of viral particles, occurs during the maturation of viral particles within DMVs [52]. Therefore, interfering with DMV formation could contribute to lower LVPs infectivity. Electron microscopy analyses confirmed that 6-AN treatment prevents DMV formation in HCV-infected Huh7 cells (cf Fig 1). Notably, restoring NAD levels by NAM addition rescues this phenotype, suggesting a link between NAD metabolism and DMV biogenesis. Nevertheless, the impact of 6-AN on DMV formation correlates with reduced viral replication, and we cannot rule out the possibility that fewer DMVs are formed simply because viral replication is decreased. However, the impact of 6-AN on glycerophospholipid metabolism suggests that inhibiting NAD biosynthesis could constitute a promising strategy to impair the replication of viruses that rely on DMV formation for their replication.

We observed that HCV is effectively inhibited by the inhibition of NAD biosynthesis by 6-AN. Interestingly, this com-pound has been shown to also inhibit HBV [15]. Specifically in that study, 6-AN has been shown to suppress the activity of the SpI, SpII, and core promoters by downregulating the transcription factor PPARα. This leads to a reduction in HBV RNA transcription and a decrease in HBsAg production. Although HBV belongs to the *Hepadnaviridae* family and utilizes replication mechanisms distinct from HCV, the involvement of PPARα is notable, given its central role in lipid metabolism in the liver. It therefore cannot be excluded that the action of 6-AN modulates the activity of nuclear factors involved in the metabolic regulation of the cell. Our analyses revealed that 6-AN treatment leads to a marked reduction in intracellular TG content, decreased secretion of low-density LVPs that have the highest infectivity. The downregula-tion of PPARα by 6-AN could thus account for the reduction in intracellular lipid droplets we observed, as well as the associated decrease in the synthesis of LVPs. Nevertheless, whether the modulation of PPARα activity contributes to

PLOS Pathogens

the decrease in HCV infectivity observed following 6-AN treatment remains to be determined. Therefore, the inhibition of NADH metabolism would impede both the replication of viral genomes by constraining cellular metabolic capacities and the secretion of LVPs by hindering the efficient loading of VLDLs with intracellular triglycerides, upon which LVP secretion relies.

Inhibiting NAD biosynthesis, particularly the salvage pathway using 6-AN or NAMPT inhibitors, significantly impairs HCV replication and disrupts lipid metabolism essential for the formation of viral replication organelles (DMVs). This disruption affects the maturation and infectivity of HCV particles. Since other viruses induce such intracellular structures for their replication, and rely on similar lipid-associated pathways, these findings highlight NAD metabolism as a key target for developing broad-spectrum host-directed antiviral strategies.

## Highlights

· **NAD(H) pathway inhibitor 6-AN rewires energy and lipid metabolism in hepatocytes**

· **HCV depends on NAD(H) metabolism to form double-membrane vesicles and replicate**

· **The production of HCV Lipo-Viro-Particles is altered by NAD(H) pathway inhibitors**

## Supporting information

**S1 Fig. (A-B) Huh7 cells were cultured for 72 h in presence of increasing concentrations of 6-AN (0, 50, 100 and 500 µM).** Intracellular ATP amounts were determined using CellTiter Glo assay (Promega) and cell proliferation after Hoechst staining of nuclei and quantification of fluorescence. Are presented means ± SEM (n = 3, one sample t-test, Bonferroni-Sidák adjusted p-value for multiple comparison to condition control, n.s. non significative, *p < 0.01). **(C)** Huh7 cells were cultured for 72 h in presence or not of 100 µM 6-AN ± 500 µM NAM, before cellular viability determination using CellTox Green Cytotoxicity Assay (Promega). Are presented means ± SEM (n = 3, one-way ANOVA for multiple comparison, n.s. non significative). **(D)** Molecular structures of NAD and Thio-NAD. Ribose in blue, adenine in green, pyrophosphate in purple, nicotinamide in black and additional thiol of thio-NAD in red.
(PDF)

**S2 Fig. Huh7 cells were infected at MOI of 1.** Seventy-two hours post-infection, total mRNA was extracted and NAMPT (**A**), NAPRT (**B**) and QPRT (**C**) expression was determined by RT-qPCR using RPL13A as housekeeping gene. Gene inductions are presented as fold of control. Are presented means ± SEM of three independent experiments. Student t test, non-significative (ns). **(D-F)** Expression levels of NAMPT, NAPRT and QPRT in liver biopsies of control and HCV infected patients (Data extracted from Boldanova T et al. study ([32]; GSE84346)).
(PDF)

**S3 Fig. Twenty-four hours after seeding, cells were treated for 72 h with 100 µM 6-AN or 100 nM FK866, with or without 500 µM NAM.** After treatment, the culture medium was removed and the cells were washed with PBS. Total NAD$^+$/NADH was measured using the high sensitivity NAD/NADH Assay (Promega). For each treatment of cells, luminescence signal was measured and expressed as percentage of the untreated control. Data are presented as the mean ± SEM of three independent experiments and were analyzed using one way ANOVA, *p < 0.05, ****p < 0.0001, ns none significative.
(PDF)

**S4 Fig. Huh7 cells were seeded 24h before infection at a MOI of 1.** Seventy-two hours post infection, culture medium was replaced and 6-AN (100 µM) or FK-866 (100 nM), supplemented or not with NAM (500 µM), were added to the cultures. After 48 h of treatment, intracellular (**A**) and extracellular (**B**) viral RNA were quantified by qPCR and specific infectivity of viral particles in supernatants was determined (**C**). Are presented means ± SEM (n = 3). A and B,

one sample t-test, *p<0.05, ns none significative. C, One-Way Anova with Sidak correction for multiple comparisons, ns none significative.
(PDF)

**S5 Fig. Huh7 cells were seeded in a p24 plate and cultured for 7 days in complete medium with 2% DMSO.** The cells were then treated for 72 hours with either 100 μM 6-AN or 100 μM 6-AN+500 μM NAM. **(A)** Intracellular and **(B)** extracellular HCV-RNA were quantified by qPCR (n=5). **(C)** Viral titer was determined on Huh7.5 cells and specific infectivity calculated as the ratio of FFU/ HCV genomes within supernatant (n=2). Are presented means±SEM, n.d.: ratio none determined due to FFU=0.
(PDF)

**S6 Fig. HepaSH cells were seeded in collagen-coated 24-well plates and cultured 4 days before infection at MOI=1.** Twenty-four hours post-infection molecules or the solvent alone (Ctrl) were added at the final concentration of 100μM 6-AN±500μM NAM. Three days post infection, cells **(A)** and culture supernatants **(B)** were harvested for RNA extraction and HCV genomes were quantified by qPCR. Are presented means±SEM of 3 biological replicates proceeded in the same experiment.
(PDF)

**S7 Fig. Huh7 cells were cultured with or without 100 μM of 6-AN, Phtalic acid or Thio-NAD. (A)** Cell proliferation was determined after Hoechst staining of nuclei and quantification of fluorescence. **(B)** Intracellular ATP amounts were determined using CellTiter Glo bioluminescent assay. Are presented means±SEM, n=3, one-way ANOVA, Bonferroni adjusted p-value for multiple comparison. ns non significative, *p<0.05, ****p<0.0001.
(PDF)

**S8 Fig. (A) Principal component analysis (PCA) of the 151 metabolites with confirmed identification (levels 1 & 2a) during LC-MS/MS metabolomic analysis.** Are compared quantities of cellular metabolites determined for Huh7 cells treated for 72h with 100μM 6-AN and control cells. Red cycles correspond to control samples whereas green triangle correspond to 6-AN treated samples. **(B)** Differentially represented identified-metabolites with a *p*-value<0.05 were submitted to Metaboanalyst server (6.0 release) for standard MSEA analysis. Enrichments ratios were computed by observed hits/ expected hits and the 22 pathways with a significative enrichment ratio (FDR<0.05), were ranked and presented from the higher ratio (top of the list). In bold are pathways related to phospholipid metabolism.
(PDF)

**S9 Fig. (A-C) Huh7 cells were infected with HCV Jc1 strain at MOI=1 and cultured for 72 h with or without 6-AN (100 μM), in the absence or presence of NAM (500 μM).** 1ml of ten-fold concentrated cell-culture supernatants were separated on iodixanol gradient (3–40%). Specific infectivity of viral particles **(A),** HCV RNA content **(B)** and FFU **(C)** were determined in each of the collected density fractions.
(PDF)

**S1 Table. Polar and semi-polar metabolites quantified by LC-MS/MS in Huh7 cells treated or not with 6-AN at 100 μM.** Are presented relative quantifications of the 151 metabolites identified with a high level of confidence (annotation level 1 and 2a) in each samples, and log2 fold changes between control and 6-AN treated cells.
(XLSX)

## Acknowledgments

We acknowledge the contribution of the Société Fédérative de Recherche Biosciences (UMS3444/CNRS, US8/Inserm, ENS de Lyon, UCBL) facilities: AniRA-ImmOs metabolic phenotyping and LYMIC-PLATIM microscopy platforms. We gratefully thank **Laurence Canaple**, **Jacques Brocard and Elodie Chatre** for technical assistance.

## Author contributions

**Conceptualization:** Laure Perrin-Cocon, Pierre-Olivier Vidalain, Olivier Diaz.

**Formal analysis:** Johan Toesca, Olivier Diaz.

**Investigation:** Johan Toesca, Marion Castell, Clémence Jacquemin, Alexandre Lalande, Julien Burlaud-Gaillard, Philippe Roingeard.

**Methodology:** Johan Toesca, Julien Burlaud-Gaillard, Philippe Roingeard.

**Project administration:** Olivier Diaz.

**Supervision:** Olivier Diaz.

**Validation:** Johan Toesca, Olivier Diaz.

**Writing – original draft:** Johan Toesca, Olivier Diaz.

**Writing – review & editing:** Johan Toesca, Eva Ogire, Julien Burlaud-Gaillard, Philippe Roingeard, Christophe Ramière, Cyrille Mathieu, Laure Perrin-Cocon, Vincent Lotteau, Pierre-Olivier Vidalain, Olivier Diaz.

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
