## [Decision Letter · Decision Letter 0]

27 Aug 2025

PPATHOGENS-D-25-01682

Nicotinamide metabolism is essential for Hepatitis C Virus replication and the production of infectious Lipo-Viro-Particles

PLOS Pathogens

Dear Dr. Diaz,

Thank you for submitting your manuscript to PLOS Pathogens. After careful consideration, we feel that it has merit but does not fully meet PLOS Pathogens's publication criteria as it currently stands. Therefore, we invite you to submit a revised version of the manuscript that addresses the points raised during the review process.

Please submit your revised manuscript within 60 days Oct 26 2025 11:59PM. If you will need more time than this to complete your revisions, please reply to this message or contact the journal office at plospathogens@plos.org. Please include the following items when submitting your revised manuscript:

We look forward to receiving your revised manuscript.

Kind regards,

Volker Lohmann, Dr.

Academic Editor

PLOS Pathogens

Ashley St. John

Section Editor

PLOS Pathogens

Sumita Bhaduri-McIntosh

Editor-in-Chief

PLOS Pathogens

orcid.org/0000-0003-2946-9497

Michael Malim

Editor-in-Chief

PLOS Pathogens

orcid.org/0000-0002-7699-2064

**Additional Editor Comments :**

While both reviewers appreciate the novelty of the study and its general interest for the virology community, they both also raise concerns about claims that are not sufficiently supported by the data. In particular, the authors need to differentiate pleiotropic effects due to cytotoxicity of the drugs, indirectly inhibiting HCV replication, from effects specifically associated with NAD metabolism. This will for sure require additional controls, as suggested by reviewer 1, and a careful revision of the claims, as pointed out by reviewer 2. In particular, the claim on specific reduction of replication organelle formation (any reduced replication will result in a reduced number of replication organelles in a replication based model) as well as the claim on specific inhibition of virus production (any reduction of RNA replication will reduce virion production) requires additional controls and further experimental evidence.

**Journal Requirements:**

https://journals.plos.org/plospathogens/s/submission-guidelines#loc-parts-of-a-submission

- ® on pages: 7, and 8

- TM on pages: 6, 8, 9, and 11.

5) We have noticed that you have uploaded Supporting Information files, but you have not included a complete list of legends. Please add a full list of legends for your Supporting Information file (Graphical Abstract) after the references list.

6) We are unable to open the following Supporting Information file: ._Graphical abstract.pdf. Please kindly revise as necessary and re-upload.

7) Some material included in your submission may be copyrighted. According to PLOSu2019s copyright policy, authors who use figures or other material (e.g., graphics, clipart, maps) from another author or copyright holder must demonstrate or obtain permission to publish this material under the Creative Commons Attribution 4.0 International (CC BY 4.0) License used by PLOS journals. Please closely review the details of PLOSu2019s copyright requirements here: PLOS Licenses and Copyright. If you need to request permissions from a copyright holder, you may use PLOS's Copyright Content Permission form.

Potential Copyright Issues:

i) We note that you indicated that "Graphics were realized with BioRender software." Please confirm whether any figures are created through BioRender. If so, please specify the figures and ensure that "Created with BioRender.com” is included in the figure(s) legend(s).

Please also confirm that you hold a Premium account and provide a pdf copy of the CC BY 4.0 Licence as provided by BioRender. For instructions on how to generate a CC BY 4.0 license for your figure, please see the guidelines here: https://help.biorender.com/hc/en-gb/articles/21282341238045-Publishing-in-open-access-resources.

If you are using the free assets from BioRender, we are unable to publish these images as they are licenced under a stricter licence than CC BY 4.0. In this case we ask you to remove the BioRender images and replace them with open source alternatives.

See these open source resources you may use to replace images / clip-art:

- https://bioart.niaid.nih.gov/

- https://bioicons.com/

- https://healthicons.org/

- https://scidraw.io/

- https://reactome.org/icon-lib

- https://www.phylopic.org/images

- https://journals.plos.org/plosbiology/article?id=10.1371/journal.pbio.3002395

8) Thank you for stating that "materials are available from the corresponding author upon reasonable request."

1. In a public repository

2. Within the manuscript itself

3. Uploaded as supplementary information.

Or you can provide more details on how readers can request these, preferably a third party, institutional email address, ideally for a data access committee or ethics committee, that readers can contact.

NOTE: Ideally, an author cannot be the point of contact for such requests.

Please also confirm at this time whether or not your submission contains all raw data required to replicate the results of your study. Authors must share the “minimal data set” for their submission. PLOS defines the minimal data set to consist of the data required to replicate all study findings reported in the article, as well as related metadata and methods (https://journals.plos.org/plosone/s/data-availability#loc-minimal-data-set-definition).

9) Please amend your detailed Financial Disclosure statement. This is published with the article. It must therefore be completed in full sentences and contain the exact wording you wish to be published.

1) State what role the funders took in the study. If the funders had no role in your study, please state: "The funders had no role in study design, data collection and analysis, decision to publish, or preparation of the manuscript.".

10) Thank you for stating "The authors have no relevant financial or non-financial interests to disclose."

Please revise your current Competing Interest statement to the standard "The authors have declared that no competing interests exist."

11) Please ensure that the funders and grant numbers match between the Financial Disclosure field and the Funding Information tab in your submission form. Note that the funders must be provided in the same order in both places as well. Currently, by the Institut National de la Santé et la Recherche Médicale (INSERM), the Centre National de Recherche Scientifique (CNRS), and Université Claude Bernard Lyon I (UCBL)" are missing from the the Funding Information tab.

**Reviewers' Comments:**

Reviewer's Responses to Questions

**Part I - Summary**

Reviewer #1: In their manuscript "Nicotinamide metabolism is essential for Hepatitis C Virus replication and the production of infectious Lipo-Viro-Particles” Toesca and collogues explore the function of NAD metabolism in HCV infection in hepatoma cells. While the results are certainly interesting for virologists there are a couple of caveats that have to be resolved prior to publication. The authors use hepatoma calls throughout their experiments and should refrain from stating that they used hepatocytes. All cancer cells including hepatoma cells are known to have a different metabolism than primary cells and human hepatocytes. Thus, the authors have to perform key experiments in hepatocytes to validate their findings. In addition, the simplest explanation that HCV replication is lower in inhibitor-treated cells due to lower ATP levels and slower cell growth needs to be addressed.

Reviewer #2: In their manuscript, Toesca et al. study how interference with the NAD synthesis pathway affects HCV replication and virus production. They find that blocking NAD synthesis, primarily through the salvage pathway, which is responsible for the main NAD biosynthesis, leads to reduced metabolism and cell proliferation. This in turn affects HCV replication, e.g. through prevention of formation of double membrane vesicles (DMVs). Lack of NAD also changes lipid metabolism in general, which may contribute to effects on HCV replication organelles and lipo-viro particle (LVP) production. The study in general is sound and the experiments well performed. I do have a few specific comments, however:

**Part II – Major Issues: Key Experiments Required for Acceptance**

Please use this section to detail the key new experiments or modifications of existing experiments that should be absolutely required to validate study conclusions.required to validate study conclusions.

Reviewer #1: 1. The main issue is that HCV infection kinetics depend on cell growth and cell density. For example, when during RNAi experiments cells growth is slightly reduced (see, e.g., PMID: 25616068). Thus, all detrimental effects observed on HCV infection could be due to the effect the inhibitors have on cell growth. The authors need address this issue e.g., by using DMSO to arrest the cell cycle and then perform infection and treatment experiments as a control.

2. Do the inhibitors that target other parts of the NAD metabolic pathways also affect cell growth and ATP levels? If cell growth and ATP levels are reduced to a similar level by all inhibitors, the effect of 6-AN is rather specific. If not the reason for lower replication rates is likely due to the reduced cell growth and ATP levels unless the authors can demonstrate that that is not the case.

3. As 6-AN (and the other inhibitors) reduces HCV replication in replicon assays all downstream effects on virus assembly and release could be secondary. The authors should assess HCV release of cells that are chronically infected and then treated with the inhibitor for a short period of time. Ideally at cell density at the time of harvesting the supernatant should be similar.

4. As explained above cancer cells have a different metabolic profile. Thus, the authors need to perform key experiments in primary human hepatocytes or hepatocytes differentiated from stem cells to confirm the relevance of NAD metabolism in a more physiological cell model.

5. Does HCV infection itself affect the enzymes involved in NAD metabolism. Are gene or protein expression levels altered or is the enzymatic activity affected?

6. Figure 1 convincingly shows that HCV replication of the replicon cells is lower but is the RLU normalized to cell number or protein content of the replicon cells? Lower cell numbers could cause lower RLUs.

7. dsRNA staining can be used as a proxy for HCV RNA replication. Please perform IF with proper quantification of dsRNA foci (preferred) or quantify the amount membranous web area over several cells in EM images to assess if RNA replication complex formation in individual cells is less efficient.

8. Oil-red-O staining in isopropanol results in lipid droplet fusion. Please repeat the experiments using BODIPY or other state of the art lipid droplet stains to avoid artifacts in the quantification and characterization of lipid droplets.

Reviewer #2: In general, it will be important for the authors to better emphasize that the effects of interfering with NAD metabolism is an indirect effect on HCV. In studies of host factors, we normally would disregard conditions where cellular metabolism and proliferation are significantly affected. I recognize that it in this case is a premise that NAD-dependent metabolism is what affects HCV infection. But this would have to more clearly stated and discussed, since the cells obviously are not in a good condition absent of NAD, which unsurprisingly affects viral replication.

On a related note, the authors state that “6-AN effectively restricts the replication of HCV in Huh7 cells without affecting cellular viability”. This needs to be rephrased in the light of 6-AN actually decreasing cell growth and metabolic activity. Although it may not disrupt membrane integrity of the cells already present, this decreased cellular metabolism clearly could (and does) explain the lowered HCV replication levels.

Also, the 100nM of FK866 (or STF-118804) used has a significant impact on cellular metabolism and proliferation. Therefore, again, this cannot be described as a direct effect on HCV infection. Instead, the clearly affected metabolic state of the cell leads to an environment less conveying for HCV replication.

Effects on virus production:

Whereas the effects on viral replication is clear, I am not convinced that 6-AN or FK866 treatment inhibits production of infectious particles as such.

Experiments on virus production (Fig. 3 and S3): Conclusions cannot be drawn as done here. It was already demonstrated that 6-AN and FK866 dramatically reduce viral replication (replicon data), including through inhibition of DVM formation. With a generally much attenuated replication level, it would obviously be expected that virus production is also highly affected. The results presented in Fig. 3 and S3 can therefore not be used to draw conclusions on any effects of inhibition of the NAD pathway on viral assembly and release. Extracellular RNA may also be released e.g. from dying cells or through exocytosis. With this in mind, the statements on specific infectivity cannot be trusted.

Conclusions on this would instead require the density centrifugation of the viral particles released, and comparison of FFUs and viral RNA titers in the resulting fractions, as presented in Fig. 7. This data can then be used to make statements about reduced specific infectivity in the low-density fractions. However, not as to whether virus production levels as such are affected. In conclusion, it is OK to keep Fig. 3 and S3, however, only with the simple notion that, as expected from the reduced replication levels, virus production is also decreased.

By extension, although perhaps obvious to analyze apolipoproteins, TGs and lipid droplets as done in Fig. 6, I would not find any associated differences to be causing the most relevant effects on HCV infection. Again, any impact on virus/LVP production would be less relevant if replication and DMV formation already is greatly impacted. This should be mentioned in the manuscript. Rather, given that lipid compositions are specifically regulated in DMVs as also discussed in this manuscript (see e.g. Bley 2020; DOI: 10.3390/ijms21082888), it would be of greater interest to investigate any role of the perturbations of lipid metabolism on DMV formation. This is something the authors should consider adding.

Effect of 6-AN treatment on metabolism: Unlike what is stated in the Results section, the levels of NAD+ and NAM were not significantly reduced after 6-AN treatment (Fig. 4A). I suppose this is quite surprising given the theoretical expectation of 6-AN influencing NAD levels? Also, this questions the assumptions drawn from interpretation of Fig. 1-2, where it was concluded that 6-AN treatment affected HCV replication through decreasing NAD levels through the salvage pathway. To address this, the authors should perform intracellular quantification of NAD levels in the 6-AN and FK866 treated cells.

**Part III – Minor Issues: Editorial and Data Presentation Modifications**

Reviewer #1: 1. Most of the supplementary figures should be moved to the primary figures, e.g., Supp 1 D,E; Sup 2, Sup 3. That data could simply be merged.

2. Why was Jc1-FLAG-E2 used for the experiments in Figure 7? The authors could just show Jc1 for consistency.

Reviewer #2: That the “DMVs are the site of viral genome replication, capsid assembly, and envelopment” is perhaps attempting to state more than we know. The authors may want to consult a newer review and qualify their statement based on current uncertainty of exactly how these processes are coupled and whether they indeed all involve the DMVs.’

I am not sure lipid droplets as such are required for HCV replication, as stated in the introduction? They clearly are for particle production.

Fig.7: In addition to the specific infectivity, it would be good to also depict the FFU titers as such across the collected densities.

PLOS authors have the option to publish the peer review history of their article (what does this mean?). If published, this will include your full peer review and any attached files.). If published, this will include your full peer review and any attached files.

.

Reviewer #1: No

Reviewer #2: No

**Figure resubmission:**
---

## [Decision Letter · Decision Letter 1]

3 Feb 2026

PPATHOGENS-D-25-01682R1

Nicotinamide metabolism is essential for Hepatitis C Virus replication and the production of infectious Lipo-Viro-Particles

PLOS Pathogens

Dear Dr. Diaz,

Thank you for submitting your manuscript to PLOS Pathogens. After careful consideration, we feel that it has merit but does not fully meet PLOS Pathogens's publication criteria as it currently stands. Therefore, we invite you to submit a revised version of the manuscript that addresses the points raised during the review process.

We look forward to receiving your revised manuscript.

Kind regards,

Volker Lohmann, Dr.

Academic Editor

PLOS Pathogens

Ashley St. John

Section Editor

PLOS Pathogens

Sumita Bhaduri-McIntosh

Editor-in-Chief

PLOS Pathogens

orcid.org/0000-0003-2946-9497

Michael Malim

Editor-in-Chief

PLOS Pathogens

orcid.org/0000-0002-7699-2064

**Additional Editor Comments:**

Both reviewers feel that the revised manuscript is improved, but also strongly recommend to include the data so far restricted to the reviewers into the paper. I fully agree, since the reader should have a chance to get a comprehensive picture.

Reviewer 1 further raises an additional important point that has not been satisfactorily addressed in the revision, that might require an additional experiment.

**Journal Requirements:**

At this stage, the following Authors/Authors require contributions: Clémence Jacquemin, and Alexandre Lalande. Please ensure that the full contributions of each author are acknowledged in the "Add/Edit/Remove Authors" section of our submission form.

3) In the online submission form, you indicated that "All data generated during this study are presented within the manuscript and materials are available from the corresponding author". All PLOS journals now require all data underlying the findings described in their manuscript to be freely available to other researchers, either

1. In a public repository

2. Within the manuscript itself

3. Uploaded as supplementary information.

**Reviewers' Comments:**

Reviewer's Responses to Questions

**Part I - Summary**

Reviewer #1: The authors addressed most issues. I recommend to include the new Supplemental Figure 2 in the Main Figures as part of Figure 1. In addition, Figures R3 and R4 should be added to the manuscript and discussed as these are important experiments that help to understand and validate their findings.

Reviewer #2: In their point-by-point letter, the authors did a good faith effort in responding to the critique from the two reviewers. I notice also that both reviewers shared the major concern of differentiating between specific effects that interfering with the NAD pathways may have on HCV replication vs. general effects caused by attenuated cellular metabolism and proliferation, as well as the concern of interpreting effects on virus production in a setting of attenuated replication.

**Part II – Major Issues: Key Experiments Required for Acceptance**

Please use this section to detail the key new experiments or modifications of existing experiments that should be absolutely required to validate study conclusions.required to validate study conclusions.

Reviewer #1: The response to Q5 if the gene or protein expression or enzymatic activity are altered in infections was not sufficiently addressed. The authors could perform simple RT-qPCRs to analyze expression levels in their infection system and, importantly, present the data in the manuscript.

Reviewer #2: Curiously, despite the extra experiments performed to address reviewer questions, the authors decided against integrating many of these data sets into the revised manuscript. I would find it imperative that most of these data sets are added to the manuscript, such that not only the reviewers, but also the readers can make their own assessment with this data available.

This would in particular include the data of Fig. R9 (Effect of 6-AN and FK866 on intracellular NAD levels). While FK866 substantially reduces NAD levels, the effect of 6-AN is surprisingly small, challenging some of the NAD related conclusions made in the manuscript throughout. Although there is an effect, and the importance of this can be discussed, it would be important that the reader at least can assess this for themselves.

In addition, I would find it important to include data from Fig. R3 (attempts to look specifically at effects on virus production), and Fig. R4 (Assessment of 6-AN in primary cells)

**Part III – Minor Issues: Editorial and Data Presentation Modifications**

Reviewer #1: x-axis labelling is missing for Figures 1 C,D, 2 A,B.

Lane 395 should refer to Figure 2C-F.

Reviewer #2: I feel less strongly about the data in Fig. R5 (Effect of infection on NAMPT abundance) and Fig. R2 (effect of other inhibitors on ATP abundance), however, since this data was generated, why not make it available?

PLOS authors have the option to publish the peer review history of their article (what does this mean?). If published, this will include your full peer review and any attached files.). If published, this will include your full peer review and any attached files.

.

Reviewer #1: No

Reviewer #2: No

**Figure resubmission:**
---

## [Editor Report · Decision Letter 2]

11 Apr 2026

Dear Prof. Diaz,

We are pleased to inform you that your manuscript 'Nicotinamide metabolism is essential for Hepatitis C Virus replication and the production of infectious Lipo-Viro-Particles' has been provisionally accepted for publication in PLOS Pathogens.

Best regards,

Volker Lohmann, Dr.

Academic Editor

PLOS Pathogens

Ashley St. John

Section Editor

PLOS Pathogens

Sumita Bhaduri-McIntosh

Editor-in-Chief

PLOS Pathogens

orcid.org/0000-0003-2946-9497

Michael Malim

Editor-in-Chief

PLOS Pathogens

orcid.org/0000-0002-7699-2064
---

## [Editor Report · Acceptance letter]

Dear Prof. Diaz,

We are delighted to inform you that your manuscript, "Nicotinamide metabolism is essential for Hepatitis C Virus replication and the production of infectious Lipo-Viro-Particles," has been formally accepted for publication in PLOS Pathogens.

Best regards,

Sumita Bhaduri-McIntosh

Editor-in-Chief

PLOS Pathogens

orcid.org/0000-0003-2946-9497

Michael Malim

Editor-in-Chief

PLOS Pathogens

orcid.org/0000-0002-7699-2064